# 🦭SEAL: Semantic-Aware Hierarchical Learning for Generalized Category Discovery

**Zhenqi He**[*]  **Yuanpei Liu**[*]  **Kai Han**[†]

Visual AI Lab, The University of Hong Kong

{zhenqi_he, ypliu0}@connect.hku.hk   kaihanx@hku.hk

## Abstract

This paper investigates the problem of Generalized Category Discovery (GCD). Given a partially labelled dataset, GCD aims to categorize all unlabelled images, regardless of whether they belong to known or unknown classes. Existing approaches typically depend on either single-level semantics or manually designed abstract hierarchies, which limit their generalizability and scalability. To address these limitations, we introduce a **SE**mantic-aware hier**A**rchical **L**earning framework (**SEAL**), guided by naturally occurring and easily accessible hierarchical structures. Within SEAL, we propose a Hierarchical Semantic-Guided Soft Contrastive Learning approach that exploits hierarchical similarity to generate informative soft negatives, addressing the limitations of conventional contrastive losses that treat all negatives equally. Furthermore, a Cross-Granularity Consistency (CGC) module is designed to align the predictions from different levels of granularity. SEAL consistently achieves state-of-the-art performance on fine-grained benchmarks, including the SSB benchmark, Oxford-Pet, and the Herbarium19 dataset, and further demonstrates generalization on coarse-grained datasets. Project page: https://visual-ai.github.io/seal/

## 1 Introduction

The field of computer vision has undergone substantial progress in various tasks, including classification [53, 28], object detection [21, 52], and segmentation [27, 30, 63]. Such advancements have largely been driven by access to large-scale, human-annotated datasets [13, 37]. However, models trained on these datasets are constrained to a closed-world paradigm, limiting their predictions to the predefined labels within the training set. In contrast, there exists a wealth of unlabelled data in the open world. To capitalize on the unlabelled data, a variety of Semi-Supervised Learning (SSL) techniques [10] have been proposed, yielding notable improvements over traditional supervised learning methods. Despite substantial success in various tasks [3, 68, 11], most existing SSL methods are designed under the closed-set assumption, wherein the training and test datasets share an identical set of classes. Category discovery, initially introduced as Novel Category Discovery (NCD) [24, 29] and later extended to Generalized Category Discovery (GCD) [57, 29], has recently emerged as a compelling open-world problem, attracting significant attention. Unlike SSL, GCD tackles the challenges where the unlabelled subset may include instances from both known and unknown classes. Its primary objective is to utilise knowledge gained from labelled data to effectively categorize all samples within the unlabelled data. Concurrently, an equivalent task named Open-world Semi-Supervised Learning (OSSL) [5] has also been introduced.

---

[*]Equal contribution.

[†]Corresponding author.

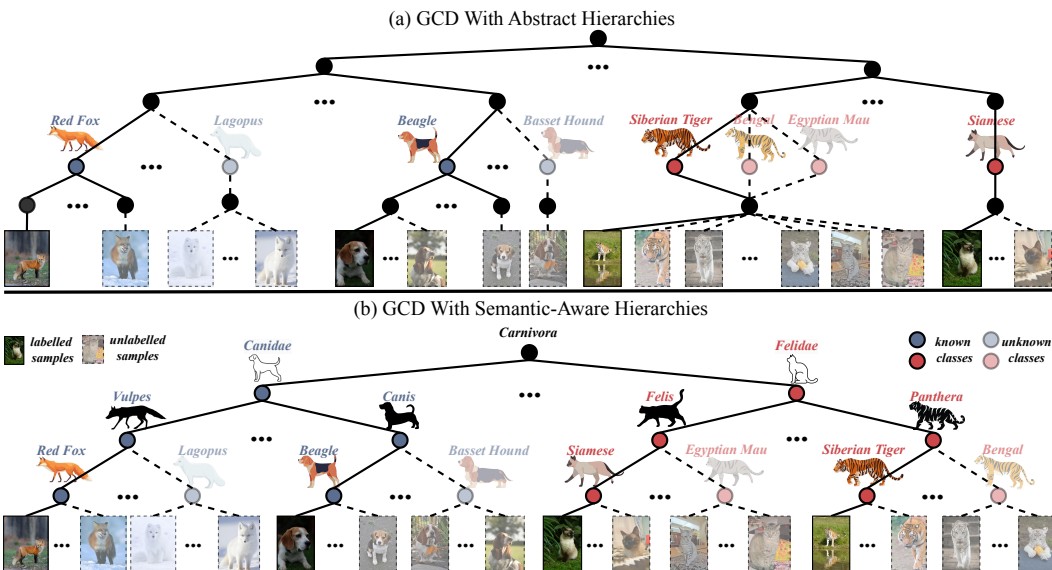

Figure 1: Comparison of SEAL and previous methods [50, 64] using hierarchical learning. (a) In previous attempts, several upper and lower levels as well as abstract concepts are defined around the ground-truth level, which may cause errors in the hierarchical structure. (b) In our method, we propose to utilise the semantic information at different levels to enhance the GCD performance.

The effectiveness of GCD is rooted in the efficient *transfer* of knowledge from known categories to *cluster* samples of both known and novel categories. As a *transfer clustering* task [24], hierarchical information has been demonstrated to be effective in GCD [50, 26] and similarly in the parallel task of OSSL [64], particularly with fine-grained datasets [58]. In [50], the hierarchical structure is composed of *abstract concepts* as an implicit binary tree, where each node represents an increasingly abstract concept derived from shared binary code prefixes, and in [51], the hierarchical structure is implicitly formed by incrementally halving the category count as the hierarchy level increases with hierarchical pseudo-labeling to provide soft supervision for the training. Similarly, CiPR [26] constructs abstract hierarchies by iteratively merging data partitions through semi-supervised clustering. The hierarchical tree in [64] consists of manually defined upper and lower levels that represent different granularities, with the number of categories per level controlled by hyperparameters. More recently, HypCD [39] implicitly models hierarchies via hyperbolic embeddings, achieving strong performance and underscoring the importance of hierarchical information for GCD. These methods build hierarchical levels from abstract, weakly supervised structures that may introduce noise and errors, ultimately affecting GCD performance. As illustrated in Fig. 1 (a), the *'Siberian Tiger'*, *'Bengal'* and *'Egyptian Mau'* can be merged into a single category while the *'Red Fox'* can be divided into multiple categories. Additionally, the high similarity among categories may result in some images of the *'Basset Hound'* being incorrectly merged with those of the *'Beagle'*. This observation naturally prompts us to consider: *whether the intrinsic, semantically grounded taxonomies present in the real world can serve as more reliable guides*. In botanical research, taxonomists commonly use labelled specimens of known species to classify newly collected, unlabelled samples into existing taxonomic hierarchies or to identify unseen species [40, 33]. Similarly, studies in closed-world visual classification [9, 65, 15] have shown that hierarchical structures enhance classification. From an information-theoretic perspective, we further deduce that such semantic-aware hierarchies yield a tighter mutual information bound, providing a principled foundation for our design.

To this end, we propose the **SE**mantic-aware hier**A**rchical **L**earning (**SEAL**) framework for GCD. Unlike previous approaches that either focus exclusively on single-granularity information [57, 59, 61, 66, 38] or rely on abstract hierarchical cues [50, 26, 64, 51, 39], SEAL effectively leverages the naturally occurring semantic hierarchies without manual design (shown in Fig. 1 (b)) and incorporates several innovative techniques tailored specifically for this task. *Firstly*, we implement a multi-task training paradigm that facilitates the simultaneous discovery of categories across several different semantic levels. *Secondly*, we introduce a Cross-Granularity Consistency (CGC)

module to align the class predictions from different granularities. *Thirdly*, we propose the Hierarchical Semantic-guided Soft Contrastive Learning to capture uncertainty in contrastive learning, ensuring that not all negative samples are treated equally. By effectively integrating these components into a cohesive framework, SEAL can be trained end-to-end in a single stage.

In summary, we make the following contributions in this paper: **(i)** We propose **SEAL**, a novel framework specifically designed to tackle the challenging GCD task by leveraging the inherent semantic hierarchies, marking the first exploration of this aspect. **(ii)** Within the SEAL framework, we develop two novel components: the Cross-Granularity Consistency (CGC) module and Hierarchical Semantic-guided Soft Contrastive Learning. These components function synergistically to significantly enhance the model's category discovery capabilities. **(iii)** Through extensive experimentation on public GCD benchmarks, SEAL consistently demonstrates its effectiveness and achieves superior performance, especially on fine-grained datasets.

## 2   Related Work

**Category Discovery.**   Novel Category Discovery (NCD) was first articulated in [24], establishing a pragmatic framework for transferring knowledge from known categories to clusters of unseen categories, framed as a transfer clustering problem. Subsequently, a variety of methods have emerged to advance the research domain [22, 23, 31, 72, 74, 18]. Generalized Category Discovery (GCD) extends the NCD framework by relaxing its assumptions, incorporating unlabelled data that features samples from both known and unknown classes [57]. Recent studies [5, 26, 48, 32, 8, 62, 38] have explored a range of strategies to tackle the challenges introduced by GCD. Notably, InfoSieve [50] and CiPR [26] guide category discovery using abstract hierarchies that are automatically inferred from the data. A similar approach is employed in the OSSL task by TIDA [64], which employs handcrafted abstract hierarchies by constructing prototypes at manually defined levels. Conversely, SimGCD [66] introduces an entropy-regularized classifier that provides a robust baseline. SPT-Net [61] builds upon SimGCD by incorporating spatial prompt tuning to emphasize salient object parts, while DebGCD [38] proposes a distribution-guided debiased learning framework to address the inherent label bias and semantic shifts in GCD.

**Hierarchical Learning.**   In the realm of hierarchical learning, numerous studies [9, 49, 65, 15] have explored the use of hierarchical label information to enhance classification performance, particularly in closed-world settings. For instance, [9] employs a multi-task framework that utilises coarse-to-fine labels to improve fine-grained recognition, whereas [71] introduces hierarchical contrastive learning to enrich representations with multi-level semantic cues. More recently, hierarchical learning has been adapted to open-set recognition, as demonstrated in [36, 67], where semantic hierarchies contribute to improved generalization to unseen classes. To the best of our knowledge, our work is the first to apply semantic-guided hierarchies to the GCD task, facilitating the effective discovery and classification of both known and novel categories.

## 3   Preliminary

**Problem Statement:** GCD aims to develop models capable of classifying unlabelled samples from known categories while simultaneously clustering those from unknown categories. Formally, we are given a labelled dataset $\mathcal{D}_l = (\boldsymbol{x}_i^l, y_i^l) \subset \mathcal{X} \times \mathcal{Y}_l$ and an unlabelled dataset $\mathcal{D}_u = (\boldsymbol{x}_i^u, y_i^u) \subset \mathcal{X} \times \mathcal{Y}_u$, where $\mathcal{Y}_l \subset \mathcal{Y}_u$. The unlabelled data includes samples from both known and novel categories. The number of known categories is denoted by $M = |\mathcal{Y}_l|$, and the total number of categories is $K = |\mathcal{Y}_l \cup \mathcal{Y}_u|$. Following prior works [23, 66, 59], we assume $K$ is known during training. When $K$ is unknown, it can be estimated using techniques such as [24, 57].

**Revisiting Baseline:** SimGCD [66] is a representative end-to-end baseline for GCD that unifies contrastive representation learning and parametric classification. The model employs a Vision Transformer [14] backbone pretrained using DINO [7], where the input image $\boldsymbol{x}_i$ is first passed through an embedding layer $\varphi$ and the feature extractor $\mathcal{F}$, followed by a projection head $\mathcal{H}$ to produce a normalized representation $\mathbf{z}_i = \mathcal{H}(\mathcal{F}(\varphi(\boldsymbol{x}_i)))/|\mathcal{H}(\mathcal{F}(\varphi(\boldsymbol{x}_i)))|$. The representation learning objective

$\mathcal{L}_{rep}$ is based on the InfoNCE loss [44]:

$$\mathcal{L}_{rep}(\boldsymbol{x}_i) = -\frac{1}{|\mathcal{P}(\boldsymbol{x}_i)|} \sum_{\mathbf{z}_i^+ \in \mathcal{P}(\boldsymbol{x}_i)} \log \sigma(\mathbf{z}_i \cdot \mathbf{z}_i^+; \tau), \tag{1}$$

where $\mathcal{P}(\boldsymbol{x}_i)$ denotes the set of positive features (*e.g.*, different views of the same image), and $\sigma(\cdot; \tau)$ is the softmax with temperature $\tau$. For labelled samples, additional positives from the same class are used to enable supervised contrastive learning.

For the parametric classification, SimGCD adopts a cosine-based classifier [20] with a learnable prototype set $\mathcal{C} = \{\boldsymbol{c}_1, ..., \boldsymbol{c}_K\}$ where each prototype $\boldsymbol{c}_k$ is $l_2$-normalized and the output probability for the $k$-th category is given by $\boldsymbol{p}_i{}^{(k)} = \sigma(\mathbf{z}_i \cdot \boldsymbol{c}_k; \tau)$. Given the pseudo-label $\boldsymbol{q}_i$ obtained from a sharpened prediction of a different view, the classification loss is:

$$\mathcal{L}_{cls}^u = \frac{1}{|\mathcal{B}|} \sum_{i \in \mathcal{B}} l_{ce}(\boldsymbol{q}_i, \boldsymbol{p}_i) - \xi H(\overline{\boldsymbol{p}}), \tag{2}$$

where $\mathcal{B}$ is current image batch and $H(\overline{\boldsymbol{p}})$ denotes the entropy of the mean prediction $\overline{\boldsymbol{p}}$. Specifically, for each $\boldsymbol{x}_i$ in the labelled batch $\mathcal{B}_l$, an additional $\boldsymbol{y}_i$ as the one-hot ground-truth vector is also used for supervised classification loss written as $\mathcal{L}_{cls}^s = \frac{1}{|\mathcal{B}_l|} \sum_{i \in \mathcal{B}_l} l_{ce}(\boldsymbol{p}_i, \boldsymbol{y}_i)$. Then, the overall classification loss is formulated as $\mathcal{L}_{cls} = (1 - \lambda_b)\mathcal{L}_{cls}^u + \lambda_b \mathcal{L}_{cls}^s$ where $\lambda_b$ is a balance factor. The final training objective combines both representation and classification terms: $\mathcal{L}_{bs} = \mathcal{L}_{cls} + \mathcal{L}_{rep}$.

## 4 Method

Before delving into methodological details, we begin with an intuitive hypothesis that underlies our framework: *Leveraging structured semantic hierarchies across multiple levels can facilitate more informative and robust feature learning for GCD setting.* To support this intuition, we first present a theoretical justification grounded in information theory, demonstrating that the incorporation of hierarchical labels yields a tighter bound on mutual information. This theoretical insight lays the foundation for the design of our approach, which we detail in the subsequent sections.

### 4.1 Theoretical Motivation

From the perspective of information theory, with denoting model parameter as $\theta$, data as $\mathcal{X}$, and label as $\mathcal{Y}$, we write $Z = f_\theta(\mathcal{X})$ as the *deterministic representation* of $\mathcal{X}$ once model $\theta$ is fixed. The optimisation objective is then to maximize the *mutual information* between $Z$ and $Y$ [4], which can be re-formulated as $\min_\theta \left\{ -I_\theta(Z_l; \mathcal{Y}_l) + \beta \left[ H_\theta(\hat{Y}_u \mid \mathcal{X}_u) - H_\theta(\hat{Y}_u) \right] \right\}$ with detailed proof provided in the Appendix, where $\hat{Y}_u$ is the model prediction for unlabelled data, and $\beta$ is the weight factor. Assuming the coarse-grained semantic hierarchical labels $\mathcal{Y}_l^{(1)}, ..., \mathcal{Y}_l^{(H-1)}$ are accessible for all labelled samples, the objective naturally extends to:

$$\min_\theta \left\{ -I_\theta(Z_l; \mathcal{Y}_l^{(1)}, \ldots, \mathcal{Y}_l^{(H)}) + \beta \left[ H_\theta(\hat{Y}_u^{(1:H)} \mid \mathcal{X}_u) - H_\theta(\hat{Y}_u^{(1:H)}) \right] \right\}. \tag{3}$$

By applying the *chain rule* of mutual information, the supervised part satisfies:

$$I_\theta(Z_l; \mathcal{Y}_l^{(1:H)}) = I_\theta(Z_l; \mathcal{Y}_l^{(H)}) + \sum_{h=1}^{H-1} I_\theta(Z_l; \mathcal{Y}_l^{(h)} \mid \mathcal{Y}_l^{(h+1)}, \ldots, \mathcal{Y}_l^{(H)}) \geq I_\theta(Z_l; \mathcal{Y}_l^{(H)}). \tag{4}$$

Analogously, for the unsupervised component, we obtain:

$$H_\theta(\hat{Y}_u^{(1:H)} \mid \mathcal{X}_u) - H_\theta(\hat{Y}_u^{(1:H)}) = -I_\theta(\mathcal{X}_u; \hat{Y}_u^{(H)}) - \sum_{h=1}^{H-1} I_\theta(\mathcal{X}_u; \hat{Y}^{(h)} \mid \hat{Y}_u^{(h+1:H)})$$

$$\leq -I_\theta(\mathcal{X}_u; \hat{Y}_u^{(H)}) = -H_\theta(\hat{Y}_u) + H_\theta(\hat{Y}_u \mid \mathcal{X}_u). \tag{5}$$

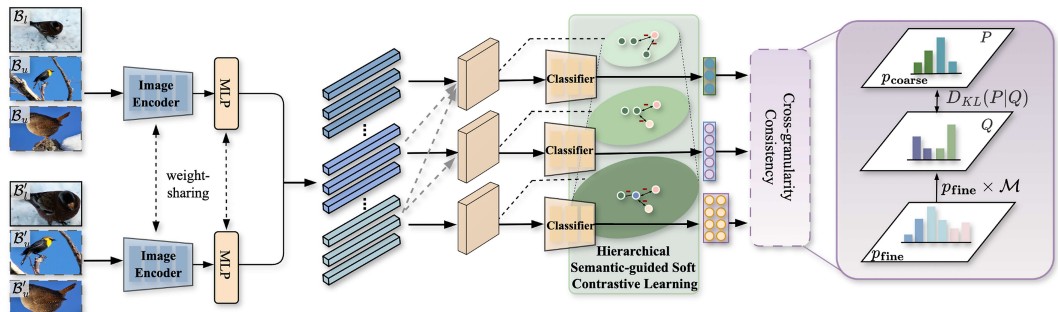

Figure 2: Overview of the proposed SEAL framework.

Combining the supervised and unsupervised parts, we have:

$$
\begin{aligned}
- I_\theta\big(Z_l; \mathcal{Y}_l^{(1)}, \ldots, \mathcal{Y}_l^{(H)}\big) &+ \beta \Big[ H_\theta\big(\hat{Y}_u^{(1:H)} \mid \mathcal{X}_u\big) - H_\theta\big(\hat{Y}_u^{(1:H)}\big) \Big] \\
&\leq - I_\theta\big(Z_l; \mathcal{Y}_l^{(H)}\big) + \beta \Big[ -H_\theta\big(\hat{Y}_u^{(H)}\big) + H_\theta\big(\hat{Y}_u^{(H)} \mid \mathcal{X}_u\big) \Big].
\end{aligned}
\tag{6}
$$

Therefore, from the perspective of information theory, incorporating semantic hierarchical labels provides a strictly tighter upper bound on the mutual information, which motivates us to introduce the semantic-guided hierarchical learning framework for GCD.

## 4.2 SEAL: Semantic-Aware Hierarchical Learning for GCD

Building on the advantages of semantic hierarchies, we propose the **SE**mantic-aware hier**A**rchical **L**earning (SEAL) framework for GCD. The overall framework is outlined in Fig. 2. In contrast to prior GCD approaches that either rely solely on single-granularity information [57, 59, 61, 66] or depend on abstract hierarchies [50, 64], we embed explicit semantic structure via three key elements: (1) a semantic-aware multi-task framework; (2) a cross-granularity consistency module to align predictions across levels; and (3) a hierarchical soft contrastive learning strategy to mitigate the "equivalent negative" assumption by weighting dissimilarity according to semantic proximity.

### 4.2.1 Semantic-aware Hierarchical Learning

We first introduce the semantic-aware multi-task framework. Inspired by [9], we advocate for a joint learning framework across multiple semantic levels, allowing information across the hierarchies to guide and strengthen representation learning at the target granularity. We define $H$ as the number of semantic levels, with corresponding ground-truth labels $\boldsymbol{y}_1, \ldots, \boldsymbol{y}_H$ ordered from coarse to fine. Our multi-task architecture couples a shared image encoder $\mathcal{F}$ followed by a projection layer $\phi$ to disentangle features for various granularities, which can be formulated as $\mathbf{z} = \phi\big(\mathcal{F}(\mathbf{x})\big) = \big[ \mathbf{z}_1; \mathbf{z}_2; \ldots; \mathbf{z}_H \big]$, where ';' denotes concatenation. Following the observation in [9] that fine-grained features can benefit coarse-grained predictions but not vice versa, we reuse lower-level features when computing coarse-level outputs. To avoid training bias towards coarse branches, we adopt a gradient controller $\Gamma$ to include fine-level features without allowing gradient backpropagation. Formally, the aggregated feature of sample $\boldsymbol{x}_i$ at the $h$-th level is $\hat{\mathbf{z}}_i = [\mathbf{z}_1; \cdots; \mathbf{z}_h; \Gamma(\mathbf{z}_{h+1}); \cdots; \Gamma(\mathbf{z}_H)]$, where $\Gamma(\cdot)$ stops gradient propagation during training. We train a GCD classifier at each level. For $h$-th level, the classification loss is denoted as $\mathcal{L}_{cls}^h$.

### 4.2.2 Cross-Granularity Consistency Self Distillation

Although multi-level classification has been widely studied in closed-world settings [9, 65], prior methods often treat each semantic level in isolation, leading to inconsistencies such as assigning labels like *'Shiba'* and *'Cat'* at different granularities for the same instance. This lack of cross-level interaction weakens the benefits of hierarchical learning. We address this with a Cross-Granularity Consistency (CGC) module that distills information between granularities to keep predictions mutually coherent. Concretely, we add a self-distillation term that minimises the KL divergence between the coarse-level posterior $p(\boldsymbol{x}_i | \boldsymbol{\theta}_h)$ and a pseudo-coarse distribution obtained by mapping the target

posterior $p(\boldsymbol{x}_i|\boldsymbol{\theta}_H)$ where $\boldsymbol{\theta}_h$ denotes the model parameters at granularity $h$. Specifically, we define a dynamic transition matrix $M_h \in \mathbb{R}^{n_H \times n_h}$ at granularity $h$ where $n_h$ denotes the number of categories at that level. Each row of $M_h$ encodes how a fine-grained class distributes over coarse classes. For known fine-grained categories, this is a fixed one-hot vector; for novel classes, we initialize with a uniform distribution and iteratively refine it during training (See Algo. 1 Dynamic Update of $M_h$). The pseudo-coarse probability thus can be computed as $p(\boldsymbol{x}_i|\boldsymbol{\theta}_H) \times M_h$ and the hierarchical consistency loss at level $h$ is defined as $D_{KL}(p(\boldsymbol{x}_i|\boldsymbol{\theta}_h)|p(\boldsymbol{x}_i|\boldsymbol{\theta}_H) \times M_h)$. Summing across levels, the overall CGC loss becomes:

$$\mathcal{L}_{cgc} = \sum_{h=1}^{H-1} D_{KL}(p(\boldsymbol{x}_i|\boldsymbol{\theta}_h)|p(\boldsymbol{x}_i|\boldsymbol{\theta}_H) \times M_h), \tag{7}$$

where $p(\boldsymbol{x}_i|\boldsymbol{\theta}_h) = \sigma(f_{\boldsymbol{\theta}_h}(\boldsymbol{x}_i), \tau_c)$ with $\sigma(\cdot)$ denoting the softmax operation and $\tau_c$ be the consistency temperature and $f_{\boldsymbol{\theta}_h}(\boldsymbol{x}_i)$ being logits computed for granularity $h$.

---

**Algorithm 1:** Dynamic Update of $M_h$

---

**Input:** Model $f$, number of class at level $h$, $n_h$, known fine-grained classes $C_{base}$
**Dynamic Update:**
Compute logits $l_h$, $l_H = f(\mathcal{D})$
Compute prediction probability $p_h$, $p_H = \sigma(l_h, \tau_c)$, $\sigma(l_H, \tau_c)$
**for** *each fine class index $k$ not in $C_{base}$* **do**
    Compute fine-grained predictions $\boldsymbol{y}_H = \operatorname{argmax}(p_H)$
    Compute average probability distribution for samples predicted as fine class $k$ :
        avg_h_prob $= \operatorname{mean}(p_h[\boldsymbol{y}_H == k])$
    Momentum update $M_h[k]$ as follows: $M_h[k] \leftarrow \lambda \cdot M_h[k] + (1-\lambda) \cdot$ avg_h_prob
    Normalize $M_h[k]$: $M_h[k] \leftarrow \dfrac{M_h[k]}{\sum M_h[k]}$

**Output:** $M_h$

---

### 4.2.3 Hierarchical Semantic-guided Soft Contrastive Learning

To strengthen the discriminative capacity of representations in GCD, we propose a Hierarchical Semantic-guided Soft Contrastive Learning approach, addressing key limitations of existing contrastive learning approaches. Prior GCD methods [23, 66, 59, 38] treat each non-positive in a mini-batch as an equally hard negative, ignoring semantic relatedness. We instead leverage the hierarchy in our multi-level framework to compute similarity-aware targets, assigning softer negative weights to semantically closer samples and preserving full penalties for unrelated ones. We compute pairwise similarities within each mini-batch at every semantic level, yielding similarity matrices $S_h$ at the $h$-th granularity, where $S_h = \frac{\mathbf{Z}_h \cdot (\mathbf{Z}_h)^\top}{\|\mathbf{Z}_h\| \times \|\mathbf{Z}_h^\top\|} \in \mathbb{R}^{B \times B}$ with $\mathbf{Z}_h$ being the features of the mini-batch at granularity $h$ and $B$ being the mini-batch size. Each fine-level matrix is then fused with its coarser counterpart, yielding a hierarchical similarity matrix $\tilde{S}_h$. We then generate semantic-aware soft labels as a matrix: $\tilde{Y}_{soft_h} = (1 - \lambda_s) \cdot \boldsymbol{I} + \lambda_s \cdot \tilde{S}_h$, where $\boldsymbol{I}$ is the identity matrix and $\lambda_s$ controls the smoothness of the semantic-aware soft labels. The resulting semantic-guided hierarchical soft contrastive loss is defined as:

$$\mathcal{L}_{hscl}^h = -\frac{1}{|B|} \sum_{i=1}^{B} \sum_{j=1}^{B} \tilde{Y}_{soft_h}(i,j) \log \frac{\exp(sim(\mathbf{z}_i, \mathbf{z}_j'))}{\sum_m^{m \neq i} \exp(sim(\mathbf{z}_i, \mathbf{z}_m'))}, \tag{8}$$

where $sim(\cdot)$ represents the similarity metric between feature $\mathbf{z}_i$ from $\boldsymbol{x}_i$ and feature $\mathbf{z}_j'$ from the augmented view of $\boldsymbol{x}_j$, and $\tilde{Y}_{soft_h}(i,j)$ refers to the $(i,j)$ element of the soft label matrix. Unlike prior works [66, 61] that rely solely on angle or distance-based measure, we adopt a hybrid metric defined as $sim(\mathbf{z}_i, \mathbf{z}_k') = \lambda_c \mathbf{z}_i \cdot \mathbf{z}_k'^\top - (1-\lambda_c) \left\| \frac{\mathbf{z}_i}{\|\mathbf{z}_i\|} - \frac{\mathbf{z}_k'}{\|\mathbf{z}_k'\|} \right\|_2$ where $\lambda_c$ is the weighting coefficient that linearly gradually decays during training. This design implements a *curriculum learning* strategy: it begins with easier angle-based cues and gradually adds distance terms to refine representations. More ablation studies about the decay schedule are in the Appendix.

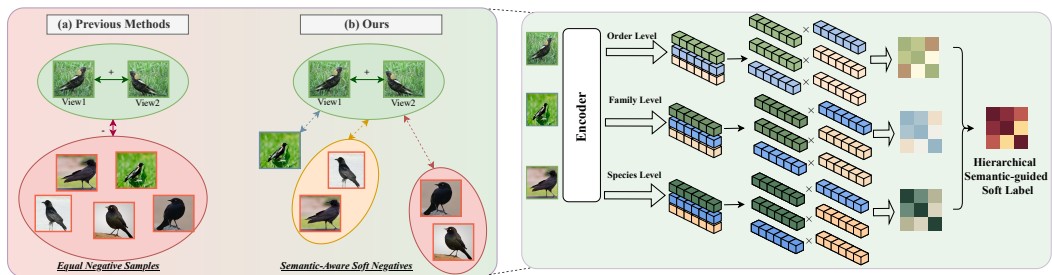

Figure 3: Overview of Hierarchical Semantic-guided Soft contrastive learning.

### 4.2.4 Overall Objective

Based on the baseline SimGCD [66] classifier, our framework is designed to be trained in a multi-task manner. We first replace the original InfoNCE loss [44] in the baseline representation loss $\mathcal{L}_{rep}$ introduced in Sec. 3 by our proposed hierarchical soft contrastive loss $\mathcal{L}_{hscl}^h$, denoting the resulting training objective at each granularity as $\mathcal{L}_{soft_{rep}}^h$. The final training objective can be formulated as

$$\mathcal{L}_{all} = \sum_h^H (\mathcal{L}_{soft_{rep}}^h + \mathcal{L}_{cls}^h) + \mathcal{L}_{cgc} \tag{9}$$

## 5 Experiment

### 5.1 Experimental Setup

**Datasets.** We conduct a comprehensive evaluation of our method across a variety of benchmarks. The main paper reports results on the Semantic Shift Benchmark (SSB) [58], which covers fine-grained datasets-CUB [60], Stanford Cars [34], and FGVC-Aircraft [42]-plus Oxford-Pet [46] and the more challenging Herbarium19 [55]. To gauge generalization on standard recognition tasks, we also include results on the generic benchmarks CIFAR-10/100[35] and ImageNet-100 [13] in the Appendix. For all datasets, we follow the class split protocol of [57], where a subset of classes is selected as the known ('Old') label set $\mathcal{Y}_l$. From these known classes, 50% of the samples are used to construct the labelled set $\mathcal{D}_l$, and the remaining images with instances from novel classes form the unlabelled set $\mathcal{D}_u$. Dataset statistics are summarized in Tab. 1.

**Evaluation metrics.** We evaluate GCD performance using clustering accuracy (*ACC*), following standard practice [57]. Specifically, given ground-truth labels $\boldsymbol{y}_i$ and predicted labels $\hat{\boldsymbol{y}}_i$ for the unlabelled set $\mathcal{D}_u$, the *ACC* is computed as:

$$ACC = \frac{1}{|\mathcal{D}_u|} \sum_{i=1}^{|\mathcal{D}_u|} \mathbb{1}(\boldsymbol{y}_i = \mathbf{h}(\hat{\boldsymbol{y}}_i)), \tag{10}$$

where $\mathbf{h}$ denotes the optimal one-to-one mapping between predicted clusters and true class labels. For a comprehensive evaluation, we report *ACC* separately for all classes ('All'), known classes ('Old'), and novel classes ('New').

Table 1: Overview of dataset, including the classes in the labelled and unlabelled sets ($|\mathcal{Y}_l|, |\mathcal{Y}_u|$) and counts of images ($|\mathcal{D}_l|, |\mathcal{D}_u|$).

| Dataset | Balance | $|\mathcal{D}_l|$ | $|\mathcal{Y}_l|$ | $|\mathcal{D}_u|$ | $|\mathcal{Y}_u|$ |
|---|---|---|---|---|---|
| CUB [60] | ✓ | 1.5K | 100 | 4.5K | 200 |
| Stanford Cars [34] | ✓ | 2.0K | 98 | 6.1K | 196 |
| FGVC-Aircraft [42] | ✓ | 1.7K | 50 | 5.0K | 100 |
| Oxford-Pet [46] | ✓ | 0.9K | 19 | 2.7K | 37 |
| Herbarium19 [55] | ✗ | 8.9K | 341 | 25.4K | 683 |

**Implementation details.** Following prior works [51, 66, 57], we adopt the ViT-B backbone [14], initialized with pretrained weights from either DINO [7] or DINOv2 [45]. The model is trained for 200 epochs using a batch size of 128 and a cosine learning rate schedule, starting from an initial learning rate of $10^{-1}$ and decaying to $10^{-4}$. All experiments are performed on a single NVIDIA L40S GPU with 24GB of memory. More details are provided in Appendix.

Table 2: Comparison of state-of-the-art GCD methods on SSB [58] benchmark. Results are reported in *ACC* across the 'All', 'Old' and 'New' categories. The highest and second-highest scores are indicated in **bold** and underline respectively.

| | Method | Venue | CUB | | | Stanford Cars | | | FGVC-Aircraft | | | Average |
|---|---|---|---|---|---|---|---|---|---|---|---|---|
| | | | All | Old | New | All | Old | New | All | Old | New | All |
| *DINOv1* | *k*-means [41] | - | 34.3 | 38.9 | 32.1 | 12.8 | 10.6 | 13.8 | 16.0 | 14.4 | 16.8 | 21.1 |
| | RankStats+ [23] | *ICLR20* | 33.3 | 51.6 | 24.2 | 28.3 | 61.8 | 12.1 | 26.9 | 36.4 | 22.2 | 29.5 |
| | UNO+ [18] | *ICCV21* | 35.1 | 49.0 | 28.1 | 35.5 | 70.5 | 18.6 | 40.3 | 56.4 | 32.2 | 37.0 |
| | ORCA [5] | *CVPR22* | 35.3 | 45.6 | 30.2 | 23.5 | 50.1 | 10.7 | 22.0 | 31.8 | 17.1 | 26.9 |
| | GCD [57] | *CVPR22* | 51.3 | 56.6 | 48.7 | 39.0 | 57.6 | 29.9 | 45.0 | 41.1 | 46.9 | 45.1 |
| | XCon [17] | *BMVC22* | 52.1 | 54.3 | 51.0 | 40.5 | 58.8 | 31.7 | 47.7 | 44.4 | 49.4 | 46.8 |
| | OpenCon [54] | *TMLR23* | 54.7 | 63.8 | 54.7 | 49.1 | 78.6 | 32.7 | - | - | - | - |
| | PromptCAL [70] | *CVPR23* | 62.9 | 64.4 | 62.1 | 50.2 | 70.1 | 40.6 | 52.2 | 52.2 | 52.3 | 55.1 |
| | DCCL [48] | *CVPR23* | 63.5 | 60.8 | 64.9 | 43.1 | 55.7 | 36.2 | - | - | - | - |
| | GPC [73] | *ICCV23* | 52.0 | 55.5 | 47.5 | 38.2 | 58.9 | 27.4 | 43.3 | 40.7 | 44.8 | 44.5 |
| | PIM [12] | *ICCV23* | 62.7 | 75.7 | 56.2 | 43.1 | 66.9 | 31.6 | - | - | - | - |
| | SimGCD [66] | *ICCV23* | 60.3 | 65.6 | 57.7 | 53.8 | 71.9 | 45.0 | 54.2 | 59.1 | 51.8 | 56.1 |
| | μGCD [59] | *NeurIPS23* | 65.7 | 68.0 | 64.6 | 56.5 | 68.1 | 50.9 | 53.8 | 55.4 | 53.0 | 58.7 |
| | InfoSieve [50] | *NeurIPS23* | **69.4** | **77.9** | **65.2** | 55.7 | 74.8 | 46.4 | 56.3 | 63.7 | 52.5 | 60.5 |
| | TIDA [64] | *NeurIPS23* | 54.7 | 72.3 | 46.2 | - | - | - | 54.6 | 61.3 | 52.1 | - |
| | CiPR [26] | *TMLR24* | 57.1 | 58.7 | 55.6 | 47.0 | 61.5 | 40.1 | - | - | - | - |
| | SPTNet [61] | *ICLR24* | 65.8 | 68.8 | 65.1 | 59.0 | 79.2 | 49.3 | 59.3 | 61.8 | 58.1 | 61.4 |
| | Yang *et al.* [69] | *ECCV24* | 61.3 | 60.8 | 62.1 | 44.3 | 58.2 | 39.1 | - | - | - | - |
| | AMEND [2] | *WACV24* | 64.9 | 75.6 | 59.6 | 52.8 | 61.8 | 48.3 | 56.4 | 73.3 | 48.2 | |
| | LegoGCD [6] | *CVPR24* | 63.8 | 71.9 | 59.8 | 57.3 | 75.7 | 48.4 | 55.0 | 61.5 | 51.7 | 58.7 |
| | MSGCD [16] | *IF25* | 63.6 | 70.7 | 60.0 | 57.7 | 75.5 | 49.9 | 56.4 | 64.1 | 52.6 | 59.2 |
| | DebGCD [38] | *ICLR25* | 66.3 | 71.8 | 63.5 | **65.3** | **81.6** | 57.4 | 61.7 | 63.9 | **60.6** | 64.4 |
| | **Ours** | - | 66.2 | 72.1 | 63.2 | **65.3** | 79.3 | 58.5 | **62.0** | 65.3 | 60.4 | **64.5** |
| *DINOv2* | *k*-means [41] | - | 67.6 | 60.6 | 71.1 | 29.4 | 24.5 | 31.8 | 18.9 | 16.9 | 19.9 | 38.6 |
| | GCD [57] | *CVPR22* | 71.9 | 71.2 | 72.3 | 65.7 | 67.8 | 64.7 | 55.4 | 47.9 | 59.2 | 64.3 |
| | CiPR [26] | *TMLR24* | **78.3** | 73.4 | **80.8** | 66.7 | 77.0 | 61.8 | 59.2 | 65.0 | 56.3 | 68.1 |
| | SimGCD [66] | *ICCV23* | 71.5 | 78.1 | 68.3 | 71.5 | 81.9 | 66.6 | 63.9 | 69.9 | 60.9 | 69.0 |
| | μGCD [59] | *NeurIPS23* | 74.0 | 75.9 | 73.1 | 76.1 | **91.0** | 68.9 | 66.3 | 68.7 | 65.1 | 72.1 |
| | SPTNet [61] | *ICLR24* | 76.3 | 79.5 | 74.6 | - | - | - | - | - | - | - |
| | DebGCD [38] | *ICLR25* | 77.5 | **80.8** | 75.8 | 75.4 | 87.7 | 69.5 | 71.9 | **76.0** | 69.8 | 74.9 |
| | **Ours** | - | 76.7 | 78.3 | 75.9 | **77.7** | 88.7 | **72.4** | **74.6** | 73.2 | **75.3** | **76.3** |

## 5.2 Main Results

We present benchmark results of our method and compare it with nineteen state-of-the-art techniques in GCD as well as three robust baselines derived from novel category discovery. All methods are based on the DINO [7] and DINOv2 [45] pre-trained backbone. This comparative evaluation encompasses performance on the fine-grained SSB benchmark [58], Oxford-Pet [46] and Herbarium19 [55], as shown in Tab. 2 and Tab. 3.

As shown in Tab. 2, our method consistently achieves state-of-the-art performance on the SSB benchmark [58] based on both DINO [7] and DINOv2 [45] pretrained backbones. Specifically, under the DINOv2 setting, our approach reaches an average 'All' accuracy of 76.3%, outperforming the previous best method, DebGCD [38], by 1.4% margin. Our framework demonstrates strong and

Table 3: Comparison with state-of-the-art GCD methods on Herbarium19 [55] and Oxford-Pet [46] on DINOv1.

| | Oxford-Pet | | | Herbarium19 | | |
|---|---|---|---|---|---|---|
| Method | All | Old | New | All | Old | New |
| *k*-means [41] | 77.1 | 70.1 | 80.7 | 13.0 | 12.2 | 13.4 |
| RankStats+ [23] | - | - | - | 27.9 | 55.8 | 12.8 |
| UNO+ [18] | - | - | - | 28.3 | 53.7 | 14.7 |
| ORCA [5] | - | - | - | 24.6 | 26.5 | 23.7 |
| GCD [57] | 80.2 | 85.1 | 77.6 | 35.4 | 51.0 | 27.0 |
| XCon [17] | 86.7 | 91.5 | 84.1 | - | - | - |
| OpenCon [54] | - | - | - | 39.3 | 58.9 | 28.6 |
| DCCL [48] | 88.1 | 88.2 | 88.0 | - | - | - |
| SimGCD [66] | 91.7 | 83.6 | 96.0 | 44.0 | 58.0 | 36.4 |
| μGCD [59] | - | - | - | 45.8 | **61.9** | 37.2 |
| InfoSieve [50] | 90.7 | **95.2** | 88.4 | 40.3 | 59.0 | 30.2 |
| DebGCD [38] | **93.0** | 86.4 | **96.5** | 44.7 | 59.4 | 36.8 |
| **Ours** | 92.9 | 88.9 | 95.0 | **46.9** | 45.8 | **48.2** |

stable improvements on both the Stanford Cars [34] and FGVC-Aircraft [42] datasets, achieving the highest accuracy under both backbone settings. This highlights the effectiveness of our semantic-guided hierarchical design and contrastive learning strategy, particularly in domains where the semantic hierarchy aligns closely with the underlying structure of man-made categories, such as vehicles and aircraft. On the CUB [60] dataset, although our method slightly lags behind DebGCD [38] and the non-parametric method InfoSieve [50], we attribute this gap to the nature of bird taxonomy based on human-annotated semantics, which may introduce inconsistencies absent in more systematically defined hierarchies like those in artificial object domains.

As shown in Tab. 3, our method achieves competitive performance on the relatively easier Oxford-Pet dataset [46], outperforming the baseline. More notably, on the more challenging Herbarium19

Table 4: Ablations. The results regarding the different components in our framework on SSB Benchmark [58]. *ACC* of 'All', 'Old' and 'New' categories are listed. Red numbers indicate the improvement over the baseline.

| | Hierarchical Learning | Consistency Self Distillation | Semantic-guided HSCL | SCars | | | CUB | | | Aircraft | | |
|---|---|---|---|---|---|---|---|---|---|---|---|---|
| | | | | All | Old | New | All | Old | New | All | Old | New |
| baseline | ✗ | ✗ | ✗ | 53.8 | 71.9 | 45.0 | 60.3 | 65.6 | 57.7 | 54.2 | 59.1 | 51.8 |
| (1) | ✓ | ✗ | ✗ | 57.5 | 67.1 | 52.9 | 57.0 | 57.8 | 56.6 | 52.8 | 56.4 | 51.0 |
| (2) | ✓ | ✓ | ✗ | 62.6 | 78.3 | 55.0 | 57.8 | 56.6 | 57.5 | 57.0 | 63.5 | 53.8 |
| (3) | ✓ | ✗ | ✓ | 64.4 | 77.5 | 58.1 | 62.5 | 67.5 | 60.0 | 57.4 | 58.5 | 56.8 |
| Ours | ✓ | ✓ | ✓ | **65.3**(+11.5) | **79.3**(+7.4) | **58.5**(+13.5) | **66.2**(+5.9) | **72.1**(+6.5) | **63.2**(+5.5) | **62.0**(+7.8) | **65.3**(+6.2) | **60.4**(+8.6) |

benchmark [55], it sets a new state-of-the-art by surpassing the previous best method, $\mu$GCD [59], by 1.1% on the 'All' accuracy. These results highlight the robustness of our approach across both simple and complex open-world discovery scenarios.

## 5.3 Analysis

**Component Analysis.** We conduct ablation studies to analyse the contributions of each major component in our framework: Hierarchical Learning, Consistency Self-Distillation, and Hierarchical Semantic-Guided Soft Contrastive Learning (HSCL). As shown in Tab. 4, we report results on the SSB benchmark [58], including Stanford Cars [34], CUB [60], and FGVC-Aircraft [42] datasets, evaluated over 'All', 'Old', and 'New' categories. Starting from the baseline trained solely with the GCD loss, we incrementally integrate the proposed components. Incorporating hierarchical learning alone (Row (1)) yields a modest improvement, particularly on the old categories. Adding consistency-based self-distillation (Row (2)) further improves alignment and stability, while semantic-guided HSCL (Row (3)) significantly boosts performance on novel classes by leveraging cross-instance semantic similarity. When all components are combined, the full framework achieves substantial gains with 11.5% on Stanford Cars, 7.8% on FGVC-Aircraft, and 5.7% on CUB.

**Hyperparameter Tuning.** In line with the practices in [66, 57], we perform hyperparameter tuning using a held-out validation split from the labelled data. Specifically, we tune the consistency temperature $\tau_c$ and the soft negative controller $\lambda_s$ based on their performance on the Stanford Cars [34] dataset. Detailed results across different hyperparameter values, evaluated on both the unlabelled training set and the validation split, are provided to assess their impact on model performance. As shown in Tab. 5, we conduct a detailed grid search over the consistency temperature $\tau_c$ and the soft negative controller $\lambda_s$ on the Stanford Cars dataset. Notably, the trends across both evaluation sets are

Table 5: Experimental results regarding consistency temperature $\tau_c$ and ratio $\lambda_s$ to control the soft negative ratio on the unlabelled set and validation set of Stanford Cars [34] dataset.

| Param. | Unlabelled Set | | | Validation Set | | |
|---|---|---|---|---|---|---|
| | All | Old | New | All | Old | New |
| $\tau_c = 0.5$ | 62.9 | 77.5 | 55.9 | 65.3 | 77.4 | 53.6 |
| $\tau_c = 0.75$ | **65.3** | 79.3 | **58.5** | **66.4** | 77.3 | **55.9** |
| $\tau_c = 1.0$ | 61.6 | 73.9 | 55.7 | 63.7 | 75.3 | 52.6 |
| $\tau_c = 1.25$ | 62.8 | **79.5** | 54.7 | 65.2 | **78.4** | 52.6 |
| $\lambda_s = 0.2$ | 63.6 | 78.9 | 56.3 | 65.6 | 78.1 | 53.5 |
| $\lambda_s = 0.4$ | 63.9 | 78.5 | 56.9 | 65.2 | 78.0 | 52.9 |
| $\lambda_s = 0.6$ | 64.7 | **80.8** | 56.9 | 66.3 | 78.3 | 54.6 |
| $\lambda_s = 0.8$ | 64.4 | 78.1 | 57.8 | 66.1 | **78.4** | 54.2 |
| $\lambda_s = 1.0$ | **65.3** | 79.3 | **58.5** | **66.4** | 77.3 | **55.9** |

highly consistent, with optimal performance achieved when $\tau_c = 0.75$ and $\lambda_s = 1.0$. These settings yield the best balance between old and new class performance, highlighting the importance of carefully tuning both the consistency strength and the soft negative ratio in our framework.

**Semantic Dimensions.** Semantic hierarchies are not restricted to a single dimension. To further demonstrate the flexibility of our framework, we additionally adopt LLM-generated labels along an alternative semantic dimension, *e.g.*, complementing the vehicle type hierarchy (SUV/Van/Coupe) with a brand-based hierarchy (Audi/BMW). Tab. 6 demonstrates that our approach achieves consistently strong performance under both semantic hierarchies. This highlights the robustness and flexibility of our proposed use of semantic-guided hierarchies across different semantic dimensions, and underscores their im-

Table 6: Results on Scars with alternative semantic hierarchies (vehicle brand vs. vehicle type) with DINOv2 pretrained backbone.

| Param. | Scars | | |
|---|---|---|---|
| | All | Old | New |
| SimGCD [66] | 71.5 | 81.9 | 66.6 |
| $\mu$GCD [59] | 76.1 | **91.0** | 68.9 |
| DebGCD [38] | 75.4 | 87.7 | 69.5 |
| SEAL(Vehicle Brand) | **77.1** | 89.0 | **71.3** |
| SEAL(Vehicle Type) | **77.7** | 88.7 | **72.4** |

portance for GCD, whether sourced from curated taxonomies, generated by LLMs, or defined along alternative semantic structures.

**Visualization.** We present a $t$-SNE [56] visualization comparing the feature representations learned by the baseline and ours. For clarity, we randomly select 20 categories, including 10 from the 'Old' set and 10 from the 'New' set. As shown in Fig. 4, our method yields tighter, better-separated clusters, indicating stronger inter-class discrimination. The zoomed view further reveals that the model preserves coarse-to-fine semantics: visually diverse subcategories within the broader *'Cab'* group lie close together, yet each remains distinct. This confirms that our method captures hierarchical structure while retaining fine-grained separability.

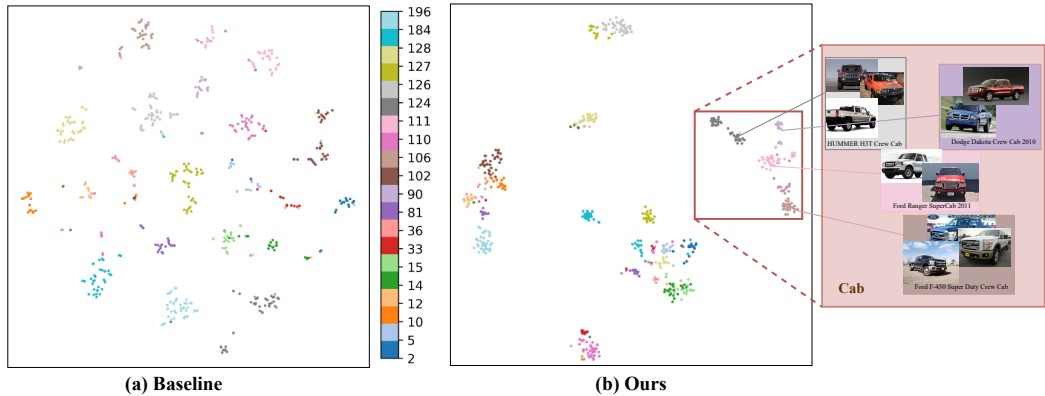

(a) Baseline         (b) Ours

Figure 4: $t$-SNE visualization of 20 classes randomly sampled from the Stanford Cars [34] dataset.

## 6 Conclusion

In this paper, we introduce a semantic-aware hierarchical learning framework for Generalized Category Discovery, composed of three key components. *Firstly*, we design a multi-task architecture that leverages naturally occurring semantic hierarchies to jointly learn coarse-to-fine category structures. *Secondly*, we propose a Cross-Granularity Consistency (CGC) module that distils information between levels, eliminating label conflicts across the hierarchy. *Thirdly*, we develop a Hierarchical Soft Contrastive Learning strategy that incorporates semantic similarity into the contrastive objective, enabling fine-grained representation learning guided by structured semantic relationships. Our framework is theoretically motivated by information-theoretic principles, which highlight the benefit of incorporating hierarchical supervision to achieve tighter theoretical bounds. Evaluations on diverse fine-grained and generic benchmarks confirm consistent, state-of-the-art gains, demonstrating both theoretical soundness and strong empirical performance.

## 7 Discussion

**Limitations.** It is important to acknowledge a limitation concerning the scale of validation within our study. The dataset used for model evaluation includes fewer than 700 instances, which constrains the breadth of category coverage. This constrained sample size may not fully represent the diversity of categories encountered in real-world scenarios. Consequently, the application of our model to category discovery in more complex and varied situations could be restricted. Further research with larger, more comprehensive datasets is warranted to validate the robustness of our findings across a wider range of categories.

**Broader Impacts.** This work presents a feasible method for discovering novel categories in unlabelled data, potentially benefiting a variety of applications such as robotics, healthcare, and autonomous driving, *etc*. However, there is a potential risk of misuse. The technology could be applied in surveillance to cluster unknown individuals, raising significant privacy concerns. Therefore, it is imperative to carefully consider ethical guidelines and legal compliance to address concerns regarding individual privacy. Additionally, to mitigate potential negative social impacts, the development of robust security protocols and systems is crucial to protect sensitive information from cyberattacks and data breaches.

## Acknowledgements

This work is supported by National Natural Science Foundation of China (Grant No. 62306251), Hong Kong Research Grant Council - Early Career Scheme (Grant No. 27208022), Hong Kong Research Grant Council - General Research Fund (Grant No. 17211024), and HKU Seed Fund for Basic Research.

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

# Appendix

# A   Theoretical Perspective

In this section, we provide an information-theoretic proof motivating the design of **SEAL**.

## A.1   Notations & Definitions

*Mutual Information* (MI) quantifies the reduction in uncertainty of one random variable given knowledge of another. We have the following definitions for MI:

**Definition.** The MI between two continuous random variables $X$ and $Y$ is formulated as.

$$I(X;Y) = \iint_{X \times Y} p_{XY}(x,y) \, \log \frac{p_{XY}(x,y)}{p_X(x) \, p_Y(y)} \, \mathrm{d}y \, \mathrm{d}x \tag{11}$$

**Definition.** The MI between two discrete random variables $X$ and $Y$ is formulated as.

$$I(X;Y) = \sum_{x,y} p_{XY}(x,y) \, \log \frac{p_{XY}(x,y)}{p_X(x) \, p_Y(y)} \tag{12}$$

The notation we used and the related formulas are given in Tab. A1

Table A1: Definition of the random variables and information measures used in this paper.

| General | |
| --- | --- |
| Labelled dataset | $\mathcal{D}_l = \{(\boldsymbol{x}_i, y_i)\}_{i=1}^n$ |
| Unlabelled dataset | $\mathcal{D}_u = \{(\boldsymbol{x}_i, y_i)\}_{i=1}^m$ |
| Image data space | $\mathcal{X}$ |
| Embedded feature space | $Z \subset \mathbb{R}^d$ |
| Label/Prediction space | $\mathcal{Y}/\hat{\mathcal{Y}} \subset \mathbb{R}^K$ |
| Euclidean distance | $D_{ij} = \|\boldsymbol{x}_i - \boldsymbol{x}_j\|_2$ |
| Cosine distance | $D_{cos_{ij}} = \frac{\boldsymbol{x}_i^\top \boldsymbol{x}_j}{\|\boldsymbol{x}_i\|\|\boldsymbol{x}_j\|}$ |
| **Model** | |
| Encoder | $f_{\boldsymbol{\theta}} : \mathcal{X} \to Z$ |
| Soft-classifier | $\mathcal{H} : \mathcal{Z} \to [0,1]^K$ |

| Random variables (RVs) | |
| --- | --- |
| Data | $X = (X_l, X_u), Y = (Y_l, Y_u)$ |
| Embedding | $Z\|\mathcal{X} \sim f_{\boldsymbol{\theta}}(\mathcal{X})$ |
| Prediction | $\hat{Y}\|Z \sim \mathcal{H}_{(}Z)$ |

| Information measures | |
| --- | --- |
| Mutual information between $Z$ and $Y$ | $\mathcal{I}(Z;Y) := \mathcal{H}(Y) - \mathcal{H}(Y\|Z)$ |
| Entropy of $Y$ | $\mathcal{H}(Y) := \mathbb{E}_{p_Y} \left[ -\log p_Y(Y) \right]$ |
| Conditional entropy of $Y$ given $Z$ | $\mathcal{H}(Y\|Z) := \mathbb{E}_{p_{YZ}} \left[ -\log p_{Y\|Z}(Y\|Z) \right]$ |
| Cross entropy (CE) between $Y$ and $\widehat{Y}$ | $\mathcal{H}(Y;\widehat{Y}) := \mathbb{E}_{p_Y} \left[ -\log p_{\widehat{Y}}(Y) \right]$ |
| Conditional CE given $Z$ | $\mathcal{H}(Y;\widehat{Y}\|Z) := \mathbb{E}_{p_{ZY}} \left[ -\log p_{\widehat{Y}\|Z}(Y\|Z) \right]$ |

## A.2   Assumptions

The following assumptions are made in our proof.

**A.1** Independent sampling between $(X_l, Y_l)$ and $(X_u, Y_u)$, which is written as $(X_l, Y_l) \perp\!\!\!\perp (X_u, Y_u)$.

**A.2** Same data distribution for $(X_l, Y_l)$ and $(X_u, Y_u)$ - the labelled and unlabelled data follow the same underlying data distribution *e.g.*, same domain.

**A.3** The representation mapping $Z = f_\theta(X)$ is deterministic and per-sample independent given parameters $\theta$.

### A.3 Theoretical Motivations

As shown in [4], from the view of information theory, the optimization objective of discriminative tasks is equivalent to maximising the MI between the learned latent features $Z$ and $Y$, which is:

$$\max_\theta I_\theta(Z; Y) \Leftrightarrow \min_\theta -I_\theta(Z; Y). \tag{13}$$

While this objective operates under the closed-world assumption, which assumes the availability of all annotations for the training data-in the GCD setting, both labelled and unlabelled data are present during training. Therefore, we further decompose the learning objective for GCD as follows. Given the *chain rule for MI:* $I(X; Y_1, \ldots, Y_n) = \sum_{i=1}^n I(X; Y_i \mid Y_{1:i-1})$, we can extend $I_\theta(Z; Y)$ as:

$$
\begin{aligned}
I_\theta(Z; Y) &= I_\theta(Z_l, Z_u; \mathcal{Y}_l, \mathcal{Y}_u) \\
&= I_\theta(Z_l, Z_u; \mathcal{Y}_l) + I_\theta(Z_l, Z_u; \mathcal{Y}_u \mid \mathcal{Y}_l) \\
&= I_\theta(Z_l; \mathcal{Y}_l) + I_\theta(Z_u | Z_l; \mathcal{Y}_l) + I_\theta(Z_l, Z_u; \mathcal{Y}_u \mid \mathcal{Y}_l) \\
&= I_\theta(Z_l; \mathcal{Y}_l) + I_\theta(Z_u | Z_l; \mathcal{Y}_l) + I_\theta(Z_l; \mathcal{Y}_u \mid \mathcal{Y}_l) + I_\theta(Z_u | Z_l; \mathcal{Y}_u \mid \mathcal{Y}_l),
\end{aligned}
\tag{14}
$$

where as $(Z_l, Y_l) \perp\!\!\!\perp (Z_u, Y_u) \iff p(z_l, y_l, z_u, y_u) = p(z_l, y_l)\, p(z_u, y_u)$, we have:

$$
\begin{aligned}
I(Z_l; \mathcal{Y}_u \mid \mathcal{Y}_l) &= \mathbb{E}_{y_l}\Big[\mathrm{KL}\big(p(z_l, y_u \mid y_l) \,\|\, p(z_l \mid y_l)\, p(y_u \mid y_l)\big)\Big] \\
&= E_{y_l}\Big[\mathrm{KL}\big(\frac{p(z_l, y_l, y_u)}{p(y_l)} \,\|\, p(z_l \mid y_l)\, p(y_u \mid y_l)\big)\Big] \\
&= E_{y_l}\Big[\mathrm{KL}\big(\frac{p(z_l, y_l)p(y_u)}{p(y_l)} \,\|\, p(z_l \mid y_l)\, p(y_u \mid y_l)\big)\Big] \\
&= E_{y_l}\Big[\mathrm{KL}\big(p(z_l \mid y_l)\, p(y_u) \,\|\, p(z_l \mid y_l)\, p(y_u \mid y_l)\big)\Big]\text{(By Bayes Rule)} \\
&= E_{y_l}\Big[\mathrm{KL}\big(p(z_l \mid y_l)\, p(y_u) \,\|\, p(z_l \mid y_l)\, p(y_u)\big)\Big]\ \text{(By Independency)} \\
&= 0.
\end{aligned}
\tag{15}
$$

As the two arguments of the KL divergence are identical, we finally have $I(Z_l; \mathcal{Y}_u | \mathcal{Y}_l) = 0$.

Similarly, for $I_\theta(Z_u | Z_l; \mathcal{Y}_l)$, we have:

$$
\begin{aligned}
I(Z_u \mid Z_l; \mathcal{Y}_l) &= I(\mathcal{Y}_l; Z_u \mid Z_l) = \mathbb{E}_{z_l}\Big[\mathrm{KL}\big(p(y_l, z_u \mid z_l) \,\|\, p(y_l \mid z_l)\, p(z_u \mid z_l)\big)\Big] \\
&= E_{z_l}\Big[\mathrm{KL}\big(\frac{p(y_l, z_l, z_u)}{p(z_l)} \,\|\, p(y_l \mid z_l)\, p(z_u \mid z_l)\big)\Big] \\
&= E_{z_l}\Big[\mathrm{KL}\big(\frac{p(y_l, z_l)p(z_u)}{p(z_l)} \,\|\, p(y_l \mid z_l)\, p(z_u \mid z_l)\big)\Big] \\
&= E_{z_l}\Big[\mathrm{KL}\big(p(y_l \mid z_l)\, p(z_u) \,\|\, p(y_l \mid z_l)\, p(z_u \mid z_l)\big)\Big]\text{(By Bayes Rule)} \\
&= E_{z_l}\Big[\mathrm{KL}\big(p(y_l \mid z_l)\, p(z_u) \,\|\, p(y_l \mid z_l)\, p(z_u)\big)\Big]\ \text{(By Independency)} \\
&= 0.
\end{aligned}
\tag{16}
$$

By the independency assumption, we have $I_\theta(Z_u | Z_l; \mathcal{Y}_u | \mathcal{Y}_l) = I(Z_u; \mathcal{Y}_u)$.

Therefore, the optimization objective is decomposed to:

$$\min_\theta -I_\theta(Z_l; \mathcal{Y}_l) - I_\theta(Z_u; \mathcal{Y}_u) \tag{17}$$

We further introduce a weight factor $\beta$ [47, 25] to balance between the supervised and unsupervised part to formulate the final objective as:

$$\min_\theta -I_\theta(Z_l; \mathcal{Y}_l) - \beta I_\theta(Z_u; \mathcal{Y}_u), \tag{18}$$

where for $I_\theta(Z_u; \mathcal{Y}_u)$, $\mathcal{Y}_u$ is unknown, we introduce a variational label distribution based on model prediction $q_\theta(\mathcal{Y}_u|X_u) \triangleq p_\theta(\hat{Y}_u|X_u)$ where $\hat{Y}$ is the softmaxed model prediction. By the data-processing inequality that information passes through a transformation, mutual information with the source cannot increase, we have $I_\theta(Z_u; \mathcal{Y}_u) \geq I_\theta(X_u; \hat{Y}_u)$. From which, we can rewrite:

$$\min -I_\theta(Z_u; Y_u) \to \min_\theta -I_\theta(X_u; \hat{Y}_u) = \min_\theta -H_\theta(\hat{\mathcal{Y}}_u) + H_\theta(\hat{\mathcal{Y}}_u \mid \mathcal{X}_u). \tag{19}$$

Thus, the overall optimization objective can be reformulated as:

$$\min_\theta -I_\theta(Z_l; \mathcal{Y}_l) + \beta\Big[-H_\theta(\hat{\mathcal{Y}}_u) + H_\theta(\hat{\mathcal{Y}}_u \mid \mathcal{X}_u)\Big], \tag{20}$$

where $\beta$ is the weight factor to balance labelled and unlabelled parts.

Assuming the coarse-grained semantic hierarchical labels $\mathcal{Y}_l^{(1)}, ..., \mathcal{Y}_l^{(H-1)}$ are accessible, the objective naturally extends to:

$$\min_\theta \Big\{ \underbrace{- I_\theta(Z_l; \mathcal{Y}_l^{(1)}, \dots, \mathcal{Y}_l^{(H)})}_{\text{supervised part}} + \underbrace{\beta\Big[H_\theta\big(\hat{Y}_u^{(1:H)} \mid \mathcal{X}_u\big) - H_\theta\big(\hat{Y}_u^{(1:H)}\big)\Big]}_{\text{unsupervised part}} \Big\}. \tag{21}$$

By applying the *chain rule*, we first decompose the supervised part as:

$$I_\theta\big(Z_l; \mathcal{Y}_l^{(1:H)}\big) = I_\theta\big(Z_l; \mathcal{Y}_l^{(H:1)}\big) = I_\theta\big(Z_l; \mathcal{Y}_l^{(H)}\big) + \sum_{h=1}^{H-1} I_\theta\big(Z_l; \mathcal{Y}_l^{(h)} \mid \mathcal{Y}_l^{(h+1)}, \dots, \mathcal{Y}_l^{(H)}\big)$$
$$\geq I_\theta\big(Z_l; \mathcal{Y}_l^{(H)}\big) \text{ as } \sum_{h=1}^{H-1} I_\theta\big(Z_l; \mathcal{Y}_l^{(h)} \mid \mathcal{Y}_l^{(h+1)}, \dots, \mathcal{Y}_l^{(H)}\big) \geq 0, \tag{22}$$

where $\sum_{h=1}^{H-1} I_\theta\big(Z_l; \mathcal{Y}_l^{(h)} \mid \mathcal{Y}_l^{(h+1)}, \dots, \mathcal{Y}_l^{(H)}\big) \geq 0$ comes from the below. For $\forall\, h \in \{1 \cdots H-1\}$, we have:

$$I_\theta\big(Z_l; \mathcal{Y}_l^{(h)} \mid \mathcal{Y}_l^{(h+1)}, \dots, \mathcal{Y}_l^{(H)}\big) =$$
$$\mathbb{E}_{c \sim p(\mathcal{Y}_l^{(h+1):(H)})}\Big[\mathrm{KL}\big(p_{Z_l, \mathcal{Y}_l^{(h)} \mid \mathcal{Y}_l^{(h+1):(H)} = c} \,\big\|\, p_{Z_l \mid \mathcal{Y}_l^{(h+1):(H)} = c}\, p_{\mathcal{Y}_l^{(h)} \mid \mathcal{Y}_l^{(h+1):(H)} = c}\big)\Big] \tag{23}$$
$$\geq 0. \text{ (Non-negativity of KL divergence)}$$

Similarly, for the unsupervised part, we can also obtain:

$$H_\theta\big(\hat{Y}_u^{(1:H)} \mid \mathcal{X}_u\big) - H_\theta\big(\hat{Y}_u^{(1:H)}\big) = -I_\theta\big(\mathcal{X}_u; \hat{Y}_u^{(H)}\big) - \sum_{h=1}^{H-1} I_\theta\big(\mathcal{X}_u; \hat{Y}^{(h)} \mid \hat{Y}_u^{(h+1:H)}\big)$$
$$\leq -I_\theta\big(\mathcal{X}_u; \hat{Y}_u^{(H)}\big) = -H_\theta\big(\hat{Y}_u\big) + H_\theta\big(\hat{Y}_u \mid \mathcal{X}_u\big). \tag{24}$$

From the above, we now have:

$$\min_\theta \Big\{ - I_\theta\big(Z_l; \mathcal{Y}_l^{(1)}, \dots, \mathcal{Y}_l^{(H)}\big) + \beta\Big[H_\theta\big(\hat{Y}_u^{(1:H)} \mid \mathcal{X}_u\big) - H_\theta\big(\hat{Y}_u^{(1:H)}\big)\Big] \Big\} \leq$$
$$\min_\theta \Big\{ - I_\theta(Z_l; \mathcal{Y}_l) + \beta\Big[-H_\theta(\hat{\mathcal{Y}}_u) + H_\theta(\hat{\mathcal{Y}}_u \mid \mathcal{X}_u)\Big] \Big\}, \tag{25}$$

where we can see that the semantic-guided hierarchies provide a tighter bound on the mutual information, which motivates us to introduce the semantic-guided hierarchical learning framework for GCD.

# B  Additional Details

## B.1  Additional Implementation Details

We adopt the class splits of labelled ('Old') and unlabelled ('New') categories in [57] for generic object recognition datasets (including CIFAR-10 [35] and CIFAR-100 [35]) and the fine-grained Semantic Shift Benchmark [58] (comprising CUB [60], Stanford Cars [34], and FGVC-Aircraft [42]), Oxford-Pet [46] and Herbarium19 [55]. Specifically, for all these datasets except CIFAR-100, $50\%$ of all classes are selected as 'Old' classes ($\mathcal{Y}_l$), while the remaining classes are treated as 'New' classes ($\mathcal{Y}_u \backslash \mathcal{Y}_l$). For CIFAR-100, $80\%$ of the classes are designated as 'Old' classes, while the remaining $20\%$ as 'New' classes. Moreover, following [57] and [66], the model's hyperparameters are chosen based on its performance on a hold-out validation set, formed by the original test splits of labelled classes in each dataset. All experiments utilize the PyTorch framework on a workstation with Nvidia L40s GPUs. The models are trained with a batch size of 128 on a single GPU for all datasets.

For the hierarchical information required by our framework, we rely exclusively on publicly available taxonomies or well-established datasets rather than any manual annotation. For the fine-grained SSB benchmarks [58], we follow the closed-world hierarchies of [9]: CUB [60] is organised into 13 orders, 38 families, and 200 species; Stanford Cars [34] is structured into 9 car types (*e.g.*, *'Cab'*, *'SUV'*) and 196 specific models; FGVC-Aircraft [42] is arranged into 30 makers (*e.g.*, *'Boeing'*, *'Douglas'*), 70 families (*e.g.*, *'Boeing 767'*), and 100 models. Oxford Pets [46] is re-cast into a two-level hierarchy with the coarse level *'Cat' vs. 'Dog'*, while Herbarium19 [55] is grouped by coarser-grained genus using the GBIF botanical database [19]. For generic benchmarks, CIFAR-10 [35] is split into the super-classes *'Vehicle'* and *'Animal'*, CIFAR-100 [35] adopts its built-in 20 super-classes, and ImageNet-100 [13] leverages the WordNet [43] taxonomy to form coarse categories. All hierarchies are obtained via public code, openly accessible biological and lexical databases or can be generated by LLMs, ensuring that our experiments reflect realistic usage without bespoke curation.

## B.2  Additional Dataset Details

Table A2: Overview of datasets we use, including the classes in the labelled and unlabelled sets ($|\mathcal{Y}_l|$, $|\mathcal{Y}_u|$) and counts of images ($|\mathcal{D}_l|$, $|\mathcal{D}_u|$). The 'FG' indicates whether the dataset is fine-grained.

| Dataset | FG | $|\mathcal{D}_l|$ | $|\mathcal{Y}_l|$ | $|\mathcal{D}_u|$ | $|\mathcal{Y}_u|$ |
|---|---|---|---|---|---|
| CIFAR-10 [35] | ✗ | 12.5K | 5 | 37.5K | 10 |
| CIFAR-100 [35] | ✗ | 20.0K | 80 | 30.0K | 100 |
| ImageNet-100 [13] | ✗ | 31.9K | 50 | 95.3K | 100 |
| CUB [60] | ✓ | 1.5K | 100 | 4.5K | 200 |
| Stanford Cars [34] | ✓ | 2.0K | 98 | 6.1K | 196 |
| FGVC-Aircraft [42] | ✓ | 1.7K | 50 | 5.0K | 100 |
| Oxford-Pet [46] | ✓ | 0.9K | 19 | 2.7K | 37 |
| Herbarium19 [55] | ✓ | 8.9K | 341 | 25.4K | 683 |

We further introduce the details of the datasets used in our paper. The statistics for the commonly used datasets are summarized in Tab. A2.

**Generic Datasets.** (1) *ImageNet*-100 [13] is a widely used dataset for natural image classification in computer vision, which is constructed by randomly subsampling 100 classes from ImageNet-1K. (2) *CIFAR*-10 & *CIFAR*-100 [35] are both natural images sized in $32 \times 32$. CIFAR-10 contains $50,000$ images spanning across 10 different classes and CIFAR-100 includes 100 classes, with each class containing 500 images.

**Fine-grained Datasets.** The most widely used fine-grained benchmark is SSB [57], which includes three datasets: CUB [60], Stanford Cars (SCars) [34], and FGVC Aircraft [42]. (1) *CUB* [60] is a widely used benchmark dataset for fine-grained visual classification tasks, particularly focused on bird species recognition. (2) *Stanford Cars* [34] is a large-scale dataset designed for fine-grained vehicle classification tasks. It contains 196 different car models, primarily spanning various makes, models, and years. (3) *FGVC-Aircraft* [42] is a fine-grained visual classification dataset focused on aircraft recognition. It contains 10,000 images spanning 100 different aircraft model variants,

with each image labelled by its corresponding model. (4) *Oxford-Pet* [46] is a large, fine-grained dataset designed for pet image classification and segmentation tasks. (5) *Herbarium19* [55] is a large-scale image collection focused on plant species identification, particularly for herbarium specimen recognition.

# C    Experiments under Realistic Situation

Following the majority of the literature, we conduct experiments mainly using the ground-truth category numbers. In this section, we test **SEAL** under more realistic conditions where neither coarse-granularity labels nor the number of classes are known. We adopt the same constraints used in earlier GCD works [57, 66]: only the *known* fine-grained classes are revealed. We first estimate the total number of targeted-granularity categories with an off-the-shelf method [57]. Next, we automatically derive coarse-level names and the fine-to-coarse mapping using ChatGPT-4o [1] with the following prompt:  `"{Targeted-grained Category Names}"` I provide these `{Number of known category} fine-grained class names, please generate the corresponding coarse-grained labels for me`.  After obtaining the coarse-granularity labels, we run the estimator [57] to infer the number of coarse categories. We test under such realistic condition for one fine-grained dataset (Stanford Cars) and one generic datasets (CIFAR100) and report the estimated class number about different granularities in Tab. A3. We compare **SEAL** with SimGCD [66], $\mu$GCD [59], and GCD [57] in Tab. A4. Even in this realistic scenario with an unknown number of categories and automatically generated coarse-granularity labels, our method outperforms existing approaches across both datasets. These results demonstrate that **SEAL** can be effectively deployed without any manual access to higher-level labels or class counts, while still achieving state-of-the-art accuracy.

Table A3: Estimated class numbers in the unlabelled data using the method proposed in [57] for both target granularity and coarse granularity.

|  | SCars (Target) | SCars (Coarse) | CIFAR-100 (Target) | CIFAR-100 (Coarse) |
|---|---|---|---|---|
| Ground-truth $K$ | 200 | 9 | 100 | 20 |
| Estimated $K$ | 231 | 9 | 100 | 20 |

Table A4: Results under the realistic scenario where neither coarse-granularity labels nor the number of classes are known. The estimated class numbers in Tab. A3 are adopted for all methods.

|  | Stanford Cars | | | CIFAR-100 | | |
|---|---|---|---|---|---|---|
| Method | All | Old | New | All | Old | New |
| GCD [57] | 35.0 | 56.0 | 24.8 | 73.0 | 76.2 | 66.5 |
| SimGCD [66] | 49.1 | 65.1 | 41.3 | 80.1 | 81.2 | 77.8 |
| $\mu$GCD [59] | 56.3 | 66.8 | 51.1 | - | - | - |
| **Ours** | **62.4** | **78.9** | **54.5** | **82.1** | **81.7** | **83.0** |

# D    Analysis on using randomly generated coarse-level labels

To further substantiate our motivation that incorrect hierarchies may introduce misleading supervision, we conduct an experiment in which the true coarse-level labels are replaced with randomly generated hierarchies. Specifically, we evaluate under two settings: $100\%$ random and $50\%$ random. As shown in Tab. A5, performance drops sharply across CUB, SCars, and Aircraft under both variants of randomly assigned hierarchical labels. This observation indicates that our gains arise from the semantic alignment of the hierarchy, not from the mere presence of a hierarchical structure.

Table A5: Ablation on randomly generated coarse-level labels.

|  | CUB | | | SCars | | | Aircraft | | |
|---|---|---|---|---|---|---|---|---|---|
|  | All | Old | New | All | Old | New | All | Old | New |
| 100% Random | 30.3 | 31.6 | 29.7 | 29.6 | 31.4 | 28.7 | 33.2 | 31.3 | 34.2 |
| 50% Random | 51.2 | 50.3 | 51.7 | 48.5 | 50.1 | 47.7 | 40.7 | 39.6 | 41.3 |
| **SEAL** | **66.2** | **72.1** | **63.2** | 65.3 | 79.3 | 58.5 | **62.0** | **65.3** | **60.4** |

# E   Results on Generic Datasets

Table A6: Comparison of state-of-the-art GCD methods on generic datasets. It includes CIFAR-10 [35], CIFAR-100 [35], ImageNet-100 [13], and the average *ACC* on All categories.

|  | Method | CIFAR-10 | | | CIFAR-100 | | | ImageNet-100 | | | Average |
|---|---|---|---|---|---|---|---|---|---|---|---|
|  |  | All | Old | New | All | Old | New | All | Old | New | All |
| *DINOv1* | *k*-means [41] | 83.6 | 85.7 | 82.5 | 52.0 | 52.2 | 50.8 | 72.7 | 75.5 | 71.3 | 69.4 |
|  | RankStats+ [23] | 46.8 | 19.2 | 60.5 | 58.2 | 77.6 | 19.3 | 37.1 | 61.6 | 24.8 | 47.4 |
|  | UNO+ [18] | 68.6 | **98.3** | 53.8 | 69.5 | 80.6 | 47.2 | 70.3 | **95.0** | 57.9 | 69.5 |
|  | ORCA [5] | 69.0 | 77.4 | 52.0 | 73.5 | **92.6** | 63.9 | 81.8 | 86.2 | 79.6 | 74.8 |
|  | GCD [57] | 91.5 | 97.9 | 88.2 | 73.0 | 76.2 | 66.5 | 74.1 | 89.8 | 66.3 | 81.1 |
|  | XCon [17] | 96.0 | 97.3 | 95.4 | 74.2 | 81.2 | 60.3 | 77.6 | 93.5 | 69.7 | 82.6 |
|  | OpenCon [54] | - | - | - | - | - | - | 84.0 | 93.8 | 81.2 | - |
|  | PromptCAL [70] | **97.9** | 96.6 | 98.5 | 81.2 | 84.2 | 75.3 | 83.1 | 92.7 | 78.3 | 87.4 |
|  | DCCL [48] | 96.3 | 96.5 | 96.9 | 75.3 | 76.8 | 70.2 | 80.5 | 90.5 | 76.2 | 84.0 |
|  | GPC [73] | 90.6 | 97.6 | 87.0 | 75.4 | 84.6 | 60.1 | 75.3 | 93.4 | 66.7 | 80.4 |
|  | SimGCD [66] | 97.1 | 95.1 | 98.1 | 80.1 | 81.2 | 77.8 | 83.0 | 93.1 | 77.9 | 86.7 |
|  | InfoSieve [50] | 94.8 | 97.7 | 93.4 | 78.3 | 82.2 | 70.5 | 80.5 | 93.8 | 73.8 | 84.5 |
|  | CiPR [26] | 97.7 | 97.5 | 97.7 | 81.5 | 82.4 | 79.7 | 80.5 | 84.9 | 78.3 | 86.6 |
|  | SPTNet [61] | 97.3 | 95.0 | **98.6** | 81.3 | 84.3 | 75.6 | 85.4 | 93.2 | 81.4 | 88.0 |
|  | DebGCD [38] | 97.2 | 94.8 | 98.4 | **83.0** | 84.6 | 79.9 | **85.9** | 94.3 | **81.6** | 88.7 |
|  | **Ours** | 97.2 | 94.7 | 98.4 | 82.1 | 81.7 | **83.0** | 84.6 | 90.9 | 81.3 | 88.0 |
| *DINOv2* | GCD [57] | 97.8 | **99.0** | 97.1 | 79.6 | 84.5 | 69.9 | 78.5 | 89.5 | 73.0 | 85.3 |
|  | CiPR [26] | **99.0** | 98.7 | 99.2 | **90.3** | 89.0 | **93.1** | 88.2 | 87.6 | 88.5 | 92.5 |
|  | SimGCD [66] | 98.7 | 96.7 | **99.7** | 88.5 | 89.2 | 87.2 | 89.9 | 95.5 | 87.1 | 92.4 |
|  | SPTNet [61] | - | - | - | - | - | - | 90.1 | 96.1 | 87.1 | - |
|  | DebGCD [38] | 98.9 | 97.5 | 99.6 | 90.1 | **90.9** | 88.6 | **93.2** | **97.0** | **91.2** | 94.1 |
|  | **Ours** | 98.9 | 98.1 | 99.3 | 89.8 | 90.4 | 89.5 | 91.3 | 93.3 | 90.3 | 93.3 |

Tab. A6 shows that **SEAL** remains effective even when only shallow hierarchies are available. With using both DINO [7] abd DINOv2 [45] pre-trained backbones, SEAL surpasses the strong SimGCD [66] baseline on all three datasets-CIFAR-10, CIFAR-100 and ImageNet-100. For generic datasets, they provide only coarse and heterogeneous groupings (*e.g.*, *'Animal' vs. 'Vehicle'* in CIFAR-10), so the hierarchy does not converge to a common parent class. By contrast, fine-grained datasets like CUB [60] share a clear taxonomic root (*e.g.*, the *class 'Aves'* for all bird species), allowing our method to exploit deeper and more coherent semantic structure. Even under this less favourable condition, **SEAL** still delivers competitive performance, confirming the robustness of our hierarchical design.

# F   Analysis on the Depth of Semantic Hierarchies

Sec. B.1 notes that our framework uses different hierarchical depths depending on dataset availability. To quantify the effect of depth, we conduct an ablation study on the two datasets that provide three explicit levels, including CUB [60] and FGVC-Aircraft [42]. For each dataset we compare: (i) a single-level baseline that uses only the target granularity, (ii) a two-level version that adds one parent level, and (iii) the full three-level setting adopted in the main paper. Tab. A7 shows that **SEAL** remains effective irrespective of the number of available semantic levels. When compared to the single-granularity baseline across both datasets, the incorporation of just one additional coarse-granularity level yields improvements of approximately 2% and 4%. These results demonstrate the robustness of our design, which can leverage richer hierarchies when they are present, while still providing significant benefits regardless of the number of hierarchies utilized.

Table A7: Ablations analysis on the depth of semantic hierarchies. *ACC* of 'All', 'Old' and 'New' categories on Stanford Cars and FGVC-Aircraft are listed.

|  | Depth of Hierarchies | Coarser Hierarchy | CUB | | | FGVC-Aircraft | | |
|---|---|---|---|---|---|---|---|---|
|  |  |  | All | Old | New | All | Old | New |
| (i) Baseline | 1 | - | 60.3 | 65.6 | 57.7 | 54.2 | 59.1 | 51.8 |
| (ii) | 2 | Family | 63.5 | 73.9 | 58.3 | 58.4 | 63.6 | 55.8 |
| (ii) | 2 | Order / Maker | 62.3 | 72.3 | 57.3 | 58.6 | 60.7 | 58.3 |
| (iii) SEAL | 3 | Order/Maker + Family | **66.2**(+5.9) | **72.1** (+6.5) | **63.2** (+5.5) | **62.0** (+7.8) | **65.3** (+6.2) | **60.4** (+8.6) |

# G   Analysis of Computational Costs

Tab. A8 presents a comprehensive analysis of the computational cost associated with our method compared to the SimGCD baseline [66] for the training stage. Despite incorporating additional multi-level supervision, our approach introduces minimal computational overhead during training. Specifically, the number of parameters increases by less than $9\%$ across all datasets (from approximately 630 MB to 688 MB), and the GFLOPs remain virtually identical, showing only a marginal increase from 17.59 to 17.60. Importantly, the training efficiency is largely preserved, with time per epoch on the unlabelled dataset increasing by no more than 0.7 seconds in all cases. These results clearly demonstrate that our framework achieves its performance improvements without sacrificing computational efficiency in training stage. At inference, however, we discard the coarse-granularity branches and keep only the classifier for the target granularity. The cost breakdown in Tab. A9 reveals that our model actually uses fewer parameters than the SimGCD baseline [66]. Runtime on the unlabelled test sets is reduced for all three datasets - Stanford Cars [34], CUB [60], and FGVC-Aircraft [42]. This economy stems from our design: a single MLP projector separates features across levels without enlarging the overall feature dimension, so the target-level head is compact at test time. Consequently, our method introduces almost no overhead during training and even lowers the computational footprint at inference, while still boosting accuracy.

Table A8: Computational cost analysis with baseline during training.

| Method | # Params (MB)↓ | | | GFLOPs↓ | | | Time per Epoch (s)↓ | | |
|---|---|---|---|---|---|---|---|---|---|
| | SCars | CUB | Aircraft | SCars | CUB | Aircraft | SCars | CUB | Aircraft |
| SimGCD [66] | 630.9 | 630.9 | 630.6 | 17.59 | 17.59 | 17.59 | 59.2 | 25.0 | 74.9 |
| Ours | 660.5 | 688.0 | 688.1 | 17.60 | 17.60 | 17.60 | 59.8 | 25.1 | 75.6 |

Table A9: Computational cost analysis with baseline at inference time.

| Method | # Params (MB)↓ | | | GFLOPs↓ | | | Time per Epoch (s)↓ | | |
|---|---|---|---|---|---|---|---|---|---|
| | SCars | CUB | Aircraft | SCars | CUB | Aircraft | SCars | CUB | Aircraft |
| SimGCD [66] | 630.9 | 630.9 | 630.6 | 17.59 | 17.59 | 17.59 | 56.9 | 34.1 | 53.1 |
| Ours | 629.0 | 627.6 | 627.5 | 17.59 | 17.59 | 17.59 | 56.8 | 34.0 | 52.9 |

# H   Analysis on the Curriculum Learning Schedule

As introduced in Sec.4.2.3, we employ a linear decay schedule for $\lambda_c$. Tab. A10 reports an ablation study on the decay strategy for the curriculum weighting coefficient $\lambda_c$. We compare fixed values ($\lambda_c = 0, 0.5, 1$), and exponential decay schedule, and our proposed SEAL with linear decay. The results consistently show that decaying schedules outperform fixed baselines, validating the effectiveness of progressively shifting focus from coarse semantic alignment to finer positional discrimination. In particular, SEAL achieves the best or comparable performance across three fine-grained datasets (CUB, Stanford-Cars, and Aircraft), demonstrating that the linear decay schedule provides a more stable and effective curriculum learning design.

Table A10: Ablation on curriculum decay strategies.

| | CUB | | | SCars | | | Aircraft | | |
|---|---|---|---|---|---|---|---|---|---|
| | All | Old | New | All | Old | New | All | Old | New |
| $\lambda_c = 0$ | 64.6 | 72.2 | 60.8 | 64.5 | 80.8 | 56.6 | 59.3 | 61.8 | 58.0 |
| $\lambda_c = 0.5$ | 65.1 | 72.8 | 61.3 | 64.1 | 77.1 | 57.8 | 58.9 | 62.5 | 57.1 |
| $\lambda_c = 1.0$ | 64.9 | 72.4 | 60.9 | 63.2 | 80.2 | 55.0 | 58.7 | 65.4 | 55.3 |
| Exp Decay | 66.1 | 71.7 | 63.3 | **66.3** | **81.2** | **60.3** | 61.2 | 63.1 | 60.2 |
| **SEAL** | **66.2** | **72.1** | **63.2** | 65.3 | 79.3 | 58.5 | **62.0** | **65.3** | **60.4** |

