# OpenReview forum: "SEAL: Semantic-Aware Hierarchical Learning for Generalized Category Discovery"
_NeurIPS.cc/2025/Conference — NeurIPS 2025 poster_

### Official Review · Reviewer_PVY1 · 2025-06-25

**Clarity:** 3
**Significance:** 3
**Originality:** 2
**Rating:** 4
**Confidence:** 3

**Summary:**

This paper presents SEAL, a semantic-aware hierarchical learning framework for GCD, which leverages natural semantic hierarchies to address the limitations of existing methods relying on single-level semantics or manual hierarchies. It introduces a Hierarchical Semantic-Guided Soft Contrastive Learning approach and a Cross-Granularity Consistency module to enhance representation learning and prediction alignment across different granularities. SEAL achieves good performance on fine-grained benchmarks.

**Questions:**

**Motivation:** How are these hierarchical data obtained? Does the method rely on predefined structures in the dataset?
**Novelty:** Multi-level semantic learning was first proposed in [41]. What is the key novelty of SEAL compared to [41]?
**Experiments:**
- Why is the improvement on CIFAR-100 and ImageNet-100 smaller than that on fine-grained datasets?
- In Table 4, why does the accuracy drop on CUB and Aircraft after applying hierarchical learning?  Similarly, why does Semantic-Guided HSCL hurt CUB but not other datasets?

**Ethical Concerns:**

["NO or VERY MINOR ethics concerns only"]

**Limitations:**

yes

**Quality:**

3

**Strengths And Weaknesses:**

**Strengths:**
- The paper is well-organized, guiding readers smoothly from motivation to methodology and experiments. Each section builds on the previous one, and visual aids like Fig. 1 help illustrate the core ideas of the hierarchy-aware design.
- Well-presented content. The proposed modules, such as Cross-Granularity Consistency (CGC) and hierarchical contrastive learning, are clearly explained. The combination of theoretical insights and implementation details makes the method easy to follow.
- Good performance. SEAL demonstrates strong performance on fine-grained datasets and consistent improvements across both known and novel categories. Ablation studies and t-SNE visualizations provide further evidence of the method’s effectiveness and the contribution of each component.

**Weaknesses:**
- The authors claim state-of-the-art performance in the abstract, but the CUB accuracy in Table 1 (66.2%) is lower than that of [41] (69.4%).
- The code has not been released.
- Missing Relevant Works: The paper overlooks important recent works, such as LegoGCD, which was accepted at CVPR 2024.

---

> ### Author Rebuttal · Authors · 2025-07-28
>
> We sincerely appreciate the reviewer's thoughtful and positive feedback. We greatly appreciate the recognition that the paper is **well-organized** from motivation to methodology and experiments, the proposed modules are **clearly explained**, and SEAL demonstrates **strong performance** across fine-grained datasets. Our detailed responses to the concerns are provided below. We‘ll also further strengthen the final manuscript based on the reviewers' feedback. A well-documented code together with all trained models will be made public.
>
> **Q1 Hierarchical Data Availability**
>
> Thanks for the valuable question. We would like to highlight that the hierarchical information used in our framework is **easily obtainable** and **naturally aligned with real-world structures**, rather than being rigid or manually imposed.
> As detailed in our supplementary material, the hierarchical labels used in our framework are either constructed based on biological taxonomies (e.g., for CUB and Herbarium), or derived from existing internal dataset structures (e.g., Aircraft, where models are already organized in the dataset by manufacturer (e.g., Boeing or Douglas) and further grouped into families such as 767 or 777.), or can be generated using large language models (LLMs) in cases where external hierarchies are unavailable.
>
> We highlight that these hierarchies are not artificially constructed, while they naturally exist in the corresponding domains and align with how taxonomic knowledge is organized in biological and real-world settings.
> For instance, in botanical research, taxonomists commonly use labelled specimens of known species to classify newly collected, unlabelled samples into existing taxonomic hierarchies or to identify unseen species [a,b]. Similarly, our method assumes access to hierarchical labels only for seen categories, which is both **practical** and **realistic**, and reflects real-world conditions where semantic structure is available for known data but not for novel categories.
>
> [a] *The role of taxonomy in species conservation* - 2004
>
> [b] *Species delimitation 4.0: integrative taxonomy meets artificial intelligence* - 2024
>
> **Q2 Distinction from [41]**
>
> Thanks for the insightful question. While Infosieve [41] introduces multi-level learning by encoding each sample using a binary path in an implicit decision tree, it inherently assumes the existence of a meaningful and fixed tree structure. However, such a structure may not accurately reflect the true semantic organization of categories, which is further supported by the observed performance gap in our experiments. For example, SEAL outperforms Infosieve by a significant margin on Scars (**65.3** vs. 55.7) and on Aircraft (**62.0** vs. 56.3).
>
> In contrast, SEAL emphasizes the use of **semantic-guided hierarchical supervision**, which is both flexible and interpretable, and does not rely on predefined or artificially imposed tree structures. Our framework is grounded in a solid information-theoretic analysis, which motivates the design and ensures that the introduced hierarchy enhances rather than misleads the learning of target-level categories. This makes SEAL more robust and better aligned with real-world semantic structures compared to tree-based approximations.
>
> **Q3 Discussion of performance on generic datasets**
>
> Thanks for the thoughtful question. The relatively smaller improvements observed on CIFAR-100 and ImageNet-100 can be attributed to the nature of their semantic structure. These datasets are **coarse-grained**, and the hierarchical relationships between classes are often **broad and loosely defined** (e.g., grouping semantically distant categories like " computer keyboard" and "clock" under a general class as "household electrical devices"), resulting in weaker and less informative supervision signals from the hierarchy.
> In contrast, fine-grained datasets such as CUB, Aircraft, and Scars feature tighter semantic groupings (e.g., species within the same genus or models from the same manufacturer), which allow our framework to more effectively exploit semantic hierarchies for representation learning and category separation. This difference naturally leads to more pronounced gains in fine-grained settings.
> We appreciate the reviewer for pointing this out, and we will further explore strategies to enhance the utility of hierarchical information in coarse-grained scenarios as part of our future work.
>
> **Q4 Ablation Studies**
>
> Thanks for the insightful question. As shown in Tab. 4 (Line 1), directly incorporating hierarchical supervision leads to performance degradation on CUB and Aircraft. While our theoretical motivation (Section 4.1) suggests that hierarchical information can be beneficial, naïvely applying it in the GCD setting can cause the training to converge to **suboptimal local minima** due to inconsistent or noisy supervision. To address this, we introduce Cross-Granularity Consistency (CGC) to ensure alignment between hierarchical levels, and Semantic-Guided Hierarchical Soft Contrastive Learning (HSCL) to better model inter-class relationships by relaxing the equally-negative assumption used in prior work. The improvements observed after applying these modules (Tab.4, *Ours*) further support their effectiveness in **robustly leveraging hierarchical signals** under the challenges of GCD.
>
> Regarding the Semantic-Guided HSCL, we clarify that it does not hurt performance on CUB. In fact, comparing Tab.4 (line 2 without HSCL) with the final _Ours_ row shows that incorporating this module improves CUB accuracy from 57.8 to __66.2__, demonstrating its effectiveness.
>
> **W1 Performance on CUB**
>
> Thanks for pointing this out. We will revise the abstract to avoid overstatement. We attribute the relatively lower performance on CUB not to a limitation of our method, but rather to the **unique data distribution** of the CUB dataset, which presents specific challenges that affect certain types of label assignment strategies.
>
> Specifically, we observe that CUB exhibits a higher degree of **intra-class multi-modality**—that is, individual classes tend to contain multiple distinct sub-clusters—compared to other datasets such as Scars and Aircraft. To empirically validate this, we conduct an analysis using Gaussian Mixture Models (GMMs) on PCA-reduced, L2-normalized features, selecting the optimal number of components M within each class via BIC (Such operation is standard for mixture selection [c], used in [d] to estimate the multimodality of a given distribution and used in class‑number estimation [e].
> Our results show that **145 out of 200** classes in CUB require $M\geq2$, indicating multi-modal structure, whereas only 7 out of 196 in SCARS and 20 out of 100 in Aircraft show such behavior. This confirms that CUB classes are often fragmented into multiple sub-modes. We report the average number of components per class and the percentage of classes with $M\geq2$.
>
> |CUB||Aircrafts||Scars||
> |-|-|-|-|-|-|
> |Avg.M|% ($M\geq2$)|Avg.M|% ($M\geq2$)|Avg.M|% ($M\geq2$)|
> |**2.07**|**72.5**|1.2|20|1.04|3.57|
>
> Under such conditions, non-parametric classifiers like k-means—as used in Infosieve [41]—are better suited to capture these sub-clusters. In contrast, our method uses a parametric classifier head trained with cross-entropy, following the widely adopted SimGCD [57] framework. While this design is effective in many settings, it may be structurally under-specified for modeling multi-modal distributions, leading to suboptimal label assignment on CUB.
>
> To further examine this, we re-evaluate our trained model using the same semi-supervised k-means in [41] clustering (averaged over 3 runs) on extracted features. Under this semi-supervised k-means setting, our method outperforms Infosieve [41] on CUB, especially when using a stronger DINOv2 backbone, which reduces initialization noise and yields more structured features. Additionally, applying semi-supervised k-means to the SimGCD baseline yields a similar performance boost, further confirming that the lower performance is not due to the learned representations, but rather due to a mismatch between classifier type and the underlying data structure.
>
> We will clarify this in the final version and include the k-means evaluation results to present a more complete picture of our method’s effectiveness.
>
> [c] Bishop, Pattern Recognition and Machine Learning, Sec. 9 (2006).
>
> [d] Wasserstein Distances, Neuronal Entanglement, and Sparsity, ICLR(2025)
>
> [e] X‑means: Extending k‑means with efficient estimation of the number of clusters, ICML(2000).
>
>
> | Method            | |             CUB                  |     |
> |------------------|-------------------------------|------|-----|
> | **DINOv2**        | **All**   | **Old**  | **New** |
> | SimGCD (Reproduce)| 72.3     | 74.1     | 71.4    |
> | SimGCD + KMeans (3runs)   | 76.8$\pm$0.4     | 76.4$\pm$1.0     | 77.1$\pm$0.3    |
> | InfoSieve (Reproduce)      | 80.7     | 84.5    | 78.8|
> | SEAL             | 76.7      | 78.3     | 75.0    |
> | SEAL + KMeans (3runs)   | **81.2$\pm$0.1**  | **79.8$\pm$0.4** | **81.9$\pm$0.3**|
>
>
> **W2 Code**
>
> Thanks for the suggestion. We will release our code upon acceptance.
>
> **W3 LegoGCD**
>
> Thanks for the suggestion. We would like to clarify that we have **cited** LegoGCD[6] and compared our method with LegoGCD[6] in Tab.2, where our method achieves stronger results.
> LegoGCD mitigates catastrophic forgetting via entropy regularization and dual-view KL divergence, whereas our method emphasizes the role of semantic hierarchies in generalized category discovery. These complementary approaches address GCD from distinct perspectives, jointly highlighting the value of structural guidance. We will include a discussion of LegoGCD in the Related Work section in the revised version.

---

> > ### Comment · Reviewer_PVY1 · 2025-08-05
> >
> > Thank you for the rebuttal. You have addressed several of my detailed questions, such as the differences from [41], the reasons for inconsistent performance on coarse- and fine-grained datasets, and some issues regarding the ablation studies. However, your response regarding the hierarchical data raises concerns. As reviewer 9SBL pointed out, this type of hierarchical data seems somewhat misaligned with real-world GCD scenarios. Therefore, I will maintain my original score.

---

> ### Author Response · Authors · 2025-08-06
>
> We sincerely appreciate your thoughtful follow-up and your acknowledgment of our clarifications regarding related work, dataset differences, and our ablation studies.
>
> Regarding hierarchical information, we further provide clarafications from two perspectives, and hope it could help.
>
> 1. Our method assumes hierarchical labels **only** for the **labelled data**, and **no hierarchy** is presumed for the **unlabelled portion**. Such hierarchies are **not manually predefined** but **derived directly** from the **labelled data**, such as existing taxonomies in biological datasets. Assuming such structure avaible for labelled data is **common** in related areas such as image classification[9, 15], and Open-Set Recognition[31, 58], and **mirrors real-world practice**: in botanical research, taxonomists commonly use **labelled specimens of known species** to **classify newly collected, unlabelled samples** into existing taxonomic hierarchies or to identify unseen species [a,b]. In this sense, our assumption **reflects a practical scenario** rather than an idealized one.
>
> 2. To evaluate more **realistic setting**, **Appendix C (Table 4)** reports results where neither the hierarchy nor the number of classes is known; we employ LLM to generate hierarchical labels from labelled data, and SEAL remains strong, outperforming baselines without ground-truth category numbers. We further provide an additional study using LLM-generated hierarchies (ChatGPT) that capture **alternative semantic dimensions**. For example, on Scars, we complement the *vehicle type* hierarchy (*e.g.*, SUV/Van/Coupe) with a *brand-based* hierarchy (*e.g.*, Audi/BMW). Under hierarchies from various semantic dimensions, our approach remains strong and achieves state-of-the-art performance, indicating that SEAL is **not tied to a single predefined hierarchy**. In this paper, we highlight that **semantic-guided hierarchies**—rather than *abstract, hand-crafted* ones—are **key for GCD**, whether sourced from curated taxonomies, generated by LLMs, or defined along different semantic dimensions. For reproducibility, we report results with the most readily accessible hierarchy in the main paper, while our framework readily **accommodates other valid hierarchies**.
>
> |Method||Scars||
> |-|-|-|-|
> ||All|Old|New|
> |SimGCD|71.5|81.9|66.6|
> |$\mu$GCD|76.1|91.0|68.9|
> |DebGCD|75.4|87.7|69.5|
> |**SEAL(Vehicle Brand)**|**77.1**|**89.0**|**71.3**|
> |**SEAL(Vehicle Type)**|**77.7**|**88.7**|**72.4**|
>
> Once again, we sincerely thank you for your thoughtful reply and for acknowledging our earlier clarifications, and we hope the additional analyses further address your concerns. If you have any further concerns, we would be happy to discuss and improve the manuscript accordingly.

---

### Official Review · Reviewer_9SBL · 2025-06-30

**Clarity:** 1
**Significance:** 3
**Originality:** 2
**Rating:** 3
**Confidence:** 4

**Summary:**

This paper addresses the Generalized Category Discovery (GCD) problem, where the goal is to categorize both known and unknown classes given a partially labeled dataset. The authors propose a new method SEAL, which leverages hierarchical structures to improve semantic representation. Concretely, the method introduces a Hierarchical Semantic-Guided Soft Contrastive Learning strategy that generates informative soft negatives and a Cross-Granularity Consistency (CGC) module to align predictions across different semantic levels. The proposed approach experiments with three fine-grained datasets.

**Questions:**

Please refer to the weaknesses.

**Ethical Concerns:**

["NO or VERY MINOR ethics concerns only"]

**Final Justification:**

I will maintain my original score, based on the following reasons:
+ The paper requires substantial revision or even rewriting, due to many unclear and ambiguous descriptions throughout. These issues make it difficult to fully understand the core ideas and assess the contributions.
+ I believe the assumption of a predefined hierarchical structure and fixed number of position categories in the GCD setting to be highly unrealistic and problematic. This setup does not align with the goals or practical constraints of real-world GCD scenarios.

**Limitations:**

The authors haven't address the limitations and potential negative societal impact of their work.

**Paper Formatting Concerns:**

N/A.

**Quality:**

2

**Strengths And Weaknesses:**

### Strengths:
+ Incorporating hierarchical information into the GCD setting is an interesting direction, as it aligns well with how semantic relationships are structured in real-world data.
+ The proposed method demonstrates sound experiment performance across three datasets.
### Weannesses:
+ The problem setting presented in this paper is not sufficiently clear, particularly in terms of the hierarchical information being leveraged. For example:
   + In Lines 54–55,  it's unclear what "naturally occurring semantic hierarchies" is supposed to refer to? Furthermore, how does the proposed setting differ from prior works such as [41, 55]?
   + The caption of Figure 1 claims that prior methods define upper and lower levels as well as abstract concepts around the ground-truth level, while the proposed method leverages semantic information at different levels. However, the figure does not clearly illustrate how the two approaches differ in their use of hierarchical information.

+ In Lines 41-43, it states that "All existing methods construct hierarchical levels based on handcrafted structures, which may introduce errors, thus affecting the GCD performance" This is a strong claim. Are there any empirical results to support this statement? This is the motivation of this paper, so I think it is necessary to provide concrete evidence that handcrafted hierarchies significantly introduce errors and affect the GCD performance.

+ In Section 4.1, the theoretical analysis assumes access to all coarse-grained hierarchical labels. However, this assumption seems unrealistic in GCD settings, where only a subset of classes is labeled.

+ It is unclear how the number of hierarchical levels $H$ is determined in Lines 152–153. Is it predefined based on the ground-truth hierarchy? Similarly, how is the number of classes per level obtained in Line 162? A more detailed explanation is necessary to clarify whether this structure is assumed given or derived from data.

+ In Lines 173–175, the description of updating the transition matrices $M_h$ for novel classes lacks clarity. The paper mentions initializing with a uniform distribution and refining during training, yet it is unclear how this refinement is implemented. While Algorithm 1 appears to be introduced for updating $M_h$, the method of obtaining $(p_h|\mathbf y_H == k)$ is not clearly explained.

+ In Lines 189–190, the process of fusing each fine-level similarity matrix with its coarser counterpart to obtain $\tilde{S}_h$ is insufficiently described. What specific operation or function is used in this fusion?

+ In Line 191, the variable $\lambda_s$ is introduced without explanation. Please define its meaning in the overall objective.

+ The paper states that the weighting coefficient $\lambda_c$ decays during training as part of a curriculum learning strategy in Lines 196-197. However, no details are provided regarding how this decay is implemented. Please clarify this component for a better understanding of the training dynamics.

---

> ### Author Rebuttal · Authors · 2025-07-29
>
> We sincerely thank the reviewer for the thoughtful and positive feedback.
> We greatly appreciate the reviewer's recognition of the **value of incorporating hierarchical information into GCD** and acknowledgment of our **sound experiment performance across three datasets**.
> Our detailed responses to mentioned concerns are provided below. We‘ll also further strengthen the final manuscript based on the reviewers' feedback. A well-documented code will be made public.
>
> ## 'Naturally occurring' & Difference with [41, 55]:
> Thanks for raising this point. In Lines 54–55, by "naturally occurring semantic hierarchies", we refer to domain-inherent structures that are **readily available** without manual design. For instance, taxonomic hierarchies in species classification (e.g., family, genus, species) are well-established and widely used in scientific databases. These reflect real-world semantic relationships and require no additional annotation. Our method leverages such **domain-native** hierarchies as semantically meaningful supervision. We will revise the wording to avoid ambiguity in the final version.
>
> Moreover, our approach differs from prior works-Infosieve [41] and TIDA [55] in both the source and use of hierarchical information. Infosieve [41] encodes category structure via binary paths in an implicit *decision tree*, assuming a fixed latent hierarchy that may not match *true semantics*—leading to suboptimal performance (e.g., SEAL outperforms Infosieve: Avg. **64.5** vs. 60.5). TIDA [55], meanwhile, uses manually defined hierarchies for prototype-level supervision, which risks introducing noisy supervision and lacks scalability. In contrast, SEAL employs readily available, **semantic-guided** hierarchies and integrates them through an **information-theoretic** framework to provide consistent, structured supervision across multiple levels, ultimately improving both representation quality and category discovery.
>
> ## Justification of motivation:
> Thanks for highlighting this point. We acknowledge that the statement in Lines 41-43 may come across as strong, and we will revise the final version to smooth our tone.
>
> To support our motivation, we provide two forms of empirical evidence illustrating the potential risks of handcrafted or semantically misaligned hierarchies:
> 1. TIDA [55] adopts manually defined **abstract hierarchies**, which, while aiming to introduce structure, may not reflect true semantic relationships. This can lead to suboptimal supervision signals. In our experiments, SEAL significantly outperforms TIDA on multiple datasets, for example, CUB (66.2 vs. 54.7) and Aircraft (62.0 vs. 54.6), suggesting that misaligned or artificial hierarchies can adversely affect GCD performance.
>
> |Method|CUB|||Aircraft|||
> |-|-|-|-|-|-|-|
> || All  | Old  | New| All   | Old  | New|
> | TIDA[55]    | 54.7  | **72.3**       | 46.2        | 54.6           | 61.3        | 52.1    |
> | SEAL  | **66.2**(+11.5)       | 72.1       | **63.2**        | **62.0** (+7.4)         | **65.3**   | **60.4**     |
>
> 2.  To further validate this, we conduct an additional experiment using **randomly generated coarse-level labels** as a substitute for semantically meaningful hierarchies. In this setting, performance **drops substantially** compared to using true semantic hierarchies, indicating that the effectiveness of our method is not simply due to hierarchy, but rather the semantic alignment of the hierarchical information.
>
> |Method|CUB||| SCars ||| Aircraft |||
> |-|-|-|-|-|-|-|-|-|-|
> ||All|Old | New|  All  | Old| New  | All| Old | New|
> | 100%  Random   | 30.3  | 31.6      | 29.7| 29.6| 31.4       | 28.7  | 33.2  | 31.3        | 34.2       |
> | 50%  Random     | 51.2 | 50.3       | 51.7 | 48.5   | 50.1  | 47.7   | 40.7  | 39.6        | 41.3        |
> | SEAL | **66.2** | **72.1** | **63.2** | **65.3** | **79.3**  | **58.5** | **62.0** | **65.3** | **60.4** |
>
> ## Assumption validity:
> Thanks for the thoughtful comment. We assume **ONLY** the availability of hierarchical labels for labelled data, which is **fully aligned** with the Generalized Category Discovery task and reflects realistic conditions in many domains.
>
> In Sec. 4.1, we leverage available hierarchical labels to decompose the mutual information via the chain rule (Line 135), showing that incorporating coarse-grained labels provides more informative and semantically structured supervision.
>
> For unlabelled data, we make no assumptions about label availability. Instead, we apply the variational lower bound:
> $I(X, Y)\geq\mathbb{E}_X \left[ \mathrm{KL}(q(Y \mid X) \parallel p(Y)) \right]$ where $q(Y \mid X)$ is approximated by the model's predictive distribution.
>
> This leads to a tractable lower bound expressed as$\mathcal{H}_\theta(\hat{Y}_u \mid X_u) - \mathcal{H}(\hat{Y}_u)$ where $\hat{Y}_u$ refers to the model's predicted label distribution for the unlabelled data, as detailed in Appendix, Lines 43–50.
>
> We will revise the manuscript to clarify this assumption and make the notation between labelled and unlabelled data more explicit in our theoretical motivation.
>
> ## Number of hierarchies:
> Thanks for the insightful question. The number of hierarchical levels $H$ is known, given the ground-truth hierarchies of labelled categories. In the main paper, we assume the number of classes at the target level is known, following common practice in the GCD community [23, 24, 33, 41, 52, 55], which ensures fair comparison under controlled settings. For the category numbers of other levels, we estimate them using the estimation method from GCD [23]. Actually, they are consistent with the ground-truth numbers (see Appendix Tab. 3). We will offer a more detailed clarification in the final version.
>
> To evaluate more realistic cases where the number of classes at the target level is also unknown, we also provide results in Appendix C (Line 110 onward), where class numbers are estimated using the method from GCD [23] with only labeled hierarchical data. As shown in Appendix Tab. 4, SEAL maintains strong performance, outperforming other baselines even without ground-truth category numbers.
>
> ## Transition Matrix Update Mechanism:
> Thanks for pointing this out. We refine the transition matrices by averaging the coarse-level model-predicted distribution ($p_h$) of samples predicted to the corresponding fine-grained class, enabling bottom-up propagation of semantic structure.
>
> Given model predictions at two granularity levels—$p_H$ (fine-grained) and $p_h$ (coarse)—we assign pseudo-labels at the fine level via $y_H = \arg\max(p_H)$ (Algo.1, line 6). For each novel class $k$ (i.e., not matching any base category), we collect samples predicted as class $k$ where $y_H == k$ and compute their average predicted distribution at the coarse level: $\text{mean}(p_h[y_H = k])$.
>
> This average captures how the fine-level class $k$ relates to higher-level categories, enabling bottom-up refinement of the hierarchy. The underlying intuition is that samples grouped under the same fine-level class should also exhibit consistent coarse-level semantics.
>
> We will improve the presentation to clarify this refinement process and its motivation.
>
> ## Fusion Mechanism & $\lambda_{s}$:
> Thanks for the insightful comment.
> In our implementation, the fused similarity matrix is computed as$\tilde{S_h} =\frac{1}{h} \sum_{n=h}^1 S_n$where $S_n$ denotes the similarity at level n. This top-down fusion integrates coarse-level semantics into fine-grained supervision.
>
> Unlike prior methods that treat all non-positive samples *equally*, we leverage hierarchical structure to impose **semantic-aware soft supervision**. For example, samples from semantically related categories (e.g., Siamese (Cat) vs. Egyptian Mau (Cat) are penalized less than those from distant ones (e.g., Siamese (Cat) vs. Beagle(Dog) ). This enables the model to learn more nuanced, structure-aware decision boundaries.
>
> The variable $\lambda_s$ in line 191 is a hyperparameter used to control the smoothness of our proposed semantic-aware soft labels.
> We further provide an additional ablation study to demonstrate the effectiveness and impact of this component. We will define its meaning in the final version.
>
> |Method|CUB||| SCars ||| Aircraft |||
> |-|-|-|-|-|-|-|-|-|-|
> ||All|Old | New|All| Old| New  | All| Old | New|
> | $\lambda_{s}=0$| 64.9| 70.4 | 59.4 | 63.8| 76.4 | 51.3|60.2|61.5|59.6|
> | SEAL| **66.2**  | **72.1** | **63.2** | **65.3** | **79.3**| **58.5** | **62.0**  | **65.3**  | **60.4** |
>
> ## Curriculum Learning Schedule:
> Thanks for the helpful comment. The weighting coefficient $\lambda_c$ follows a **linear decay schedule**: $\lambda_c = 1 - \frac{\text{epoch}}{\text{TotalEpochs}}$, gradually shifting focus from semantic alignment (via cosine similarity) to finer positional discrimination as training progresses.
>
> An ablation study comparing fixed values ($\lambda_c = 0, 0.5, 1$) and exponential decay($1 - \exp\left(-\frac{\text{epoch}}{\text{TotalEpochs}} \right)$) confirms that decaying schedules consistently outperform fixed baselines, validating the effectiveness of this curriculum design.
> |Method|CUB||| SCars ||| Aircraft |||
> |-|-|-|-|-|-|-|-|-|-|
> ||All|Old| New|All| Old| New| All| Old | New|
> | $\lambda_{c}=0$ |64.6| 72.2| 60.8 | 64.5|80.8|56.6|59.3|61.8|58.0|
> | $\lambda_{c}=0.5$| 65.1| 72.8|61.3| 64.1|77.1| 57.8  |58.9|62.5|57.1|
> | $\lambda_{c}=1$| 64.9| 72.4  | 60.9|63.2|80.2| 55.0  |58.7|65.4 |55.3|
> | Exp Declay|66.1| 71.7| 63.3 | **66.3**| **81.2**|**60.3**|61.2| 63.1 | 60.2|
> | SEAL| **66.2**| **72.1**| **63.2**|65.3| 79.3| 58.5|**62.0**| **65.3**| **60.4**|
>
> We will provide more clarification regarding this strategy in the final version.

---

> > ### Comment · Reviewer_9SBL · 2025-08-05
> >
> > I will maintain my original score, based on the following reasons:
> > + The paper requires substantial revision or even rewriting, due to many unclear and ambiguous descriptions throughout. These issues make it difficult to fully understand the core ideas and assess the contributions.
> > + I believe the assumption of a predefined hierarchical structure and fixed number of position categories in the GCD setting to be highly unrealistic and problematic. This setup does not align with the goals or practical constraints of real-world GCD scenarios.

---

> > > ### Author Response · Authors · 2025-08-05
> > > **Clarification on Hierarchy Derivation and Class-Count Assumptions**
> > >
> > > Thank you for your follow-up comments.
> > >
> > > We would like to clarify that the hierarchical structure in our method is **not manually predefined** but **derived directly from the labelled data**, such as existing taxonomies in biological datasets, as noted in Section 1 and our earlier comment. This approach aligns with real-world practice. In biological sciences, hierarchical classification arises naturally and serves as a foundational principle in taxonomy. For example, in botanical research, taxonomists commonly use labelled specimens of known species to classify newly collected, unlabelled samples into existing taxonomic hierarchies or to identify novel species [a,b].
> > >
> > > Regarding the number of categories, we only assume **a fixed number at the target (finest) level**,  in line with **common GCD practice** ( *e.g.,* **GCD [48] (CVPR’22), SimGCD [57] (ICCV’23), TIDA [55] (NeurIPS’23), InfoSieve [41] (NeurIPS’23),  SPTNet [52] (ICLR’24), LegoGCD [6] (CVPR’24), and DebGCD [33] (ICLR’25))**) to ensure fair comparison. To address more **realistic scenarios**, we further report results in **Appendix C (Table 4)** where neither the hierarchy nor the number of classes is known, estimating class counts with the existing method, and show that SEAL still maintains strong performance, outperforming baselines even without ground-truth category numbers.
> > >
> > > We appreciate your feedback and hope these clarifications are helpful. Please let us know if any further questions arise, we would be happy to discuss them.
> > >
> > > [a] The role of taxonomy in species conservation - 2004
> > >
> > > [b] Species delimitation 4.0: integrative taxonomy meets artificial intelligence - 2024

---

### Official Review · Reviewer_mwGr · 2025-07-02

**Clarity:** 3
**Significance:** 2
**Originality:** 2
**Rating:** 4
**Confidence:** 3

**Summary:**

This paper introduces SEAL, a new framework for Generalized Category Discovery. The core problem is to classify images from a dataset where some classes are known (labeled) and others are novel (unlabeled). The authors argue that existing methods are limited by their reliance on single-level semantics or manually created hierarchies. To address this, SEAL leverages naturally occurring semantic hierarchies to guide the learning process. The framework has two key components: 1) a Hierarchical Semantic-Guided Soft Contrastive Learning loss, which uses the hierarchy to select more informative negative samples, and 2) a Cross-Granularity Consistency module, which ensures that predictions are consistent across different levels of the semantic hierarchy (e.g., fine-grained classes and their coarse-grained superclasses). The method is evaluated on several fine-grained and coarse-grained benchmarks, where it reportedly achieves state-of-the-art results.

**Questions:**

1. Could you elaborate on the relationship between SEAL and DebGCD? Given their competitive performance in Table 2, a more direct comparison of their architectural and methodological differences would be helpful to understand the trade-offs and the specific scenarios where SEAL offers an advantage.
2. What is the potential explanation for the lower performance of SEAL on the CUB dataset? Does this suggest a limitation of the automatically-derived semantic hierarchy for this particular data distribution, or are there other aspects of the framework that may be less effective here?
3. Regarding the related work, could you provide a more detailed differentiation from the key methods cited in lines 75-85? Specifically, for those that also employ contrastive learning, what is the key advantage of your hierarchical semantic-guided approach?

**Ethical Concerns:**

["NO or VERY MINOR ethics concerns only"]

**Final Justification:**

The authors have successfully addressed all my concerns, including new experiments and explanations.

**Limitations:**

The authors provide a limitations section, but it is located in the appendix. According to the NeurIPS Call for Papers and reviewing guidelines, which emphasize the importance of transparency, a discussion of limitations is a key part of a strong submission. The most relevant limitations should be summarized in the main paper to provide readers with a more balanced and complete understanding of the work's scope and applicability.

**Paper Formatting Concerns:**

Minor: Table 3 appears before table 2 in the paper.

**Quality:**

2

**Strengths And Weaknesses:**

Strengths

The paper is generally well-written and easy to follow. The motivation for using semantic hierarchies in GCD is clear and compelling, and the proposed SEAL framework is described in sufficient detail.

The authors provide a theoretical motivation for their approach in the supplementary material, grounding their method in information-theoretic principles.

The supplementary material includes a valuable analysis of the computational cost of SEAL compared to baseline methods, showing consideration for the practical aspects of the proposed framework.



Weaknesses

The related work section could be strengthened. In several instances (e.g., lines 75, 77, 85), multiple prior works are cited as a group without a clear explanation of their individual contributions or how SEAL specifically improves upon them. This makes it difficult to precisely understand the originality and significance of the proposed method in the context of the literature.

The empirical results, while strong on some datasets, raise questions that are not fully addressed in the paper:

1. In table 2, the improvement in DINO (average) wrt to the second best method is only 0.1 (DebGCD). Indeed, DebGCD matches or outperforms the proposed method in 2 of the 3 datasets for All classes. The results for this backbone are not discussed, and an explanation for this difference is not given.
2. The performance on the CUB dataset is a notable concern. Across the "All," "New," and "Known" class settings, SEAL consistently lags behind methods like DebGCD and SimGCD. This underperformance on a key fine-grained benchmark is not discussed.
3. It is not immediately clear from the main text which backbone was used for the results in Table 3 (DINO or DINOv2?). Why only one for these two datasets?

---

> ### Author Rebuttal · Authors · 2025-07-29
>
> We sincerely thank the reviewer for the thoughtful feedback.
> We greatly appreciate the reviewer's recognition of the **paper's clarity**, the **clear and compelling motivation for using semantic hierarchies in GCD**, and the **detailed presentation** of the SEAL framework. We're also glad that the **theoretical justification** and **computational analysis** in the supplementary material were found to be **valuable**. Our detailed responses are provided below. We will also further strengthen the final manuscript based on the reviewers' concluding feedback. A well-documented code together with all trained models will be made public.
>
> **Q1&W1 Difference between DebGCD**
>
> Thanks for the valuable question. DebGCD addresses general *label bias*, while SEAL targets the core challenge of leveraging **semantic hierarchy** in GCD, leading to more **scalable** and **lightweight** solution, especially under **strong** representation.
>
> DebGCD tackles the *label bias* issue by introducing an auxiliary debiased classifier trained on pseudo-labels, along with an OOD-aware regularization strategy. This approach is particularly effective under weaker backbones such as DINO, where label bias is more severe and novel-class separation is more difficult. By explicitly mitigating overfitting to known classes, DebGCD improves robustness in such *less discriminative* feature spaces.
>
> In contrast, SEAL focuses on a space-agnostic challenge: how to effectively leverage **inherent semantic hierarchies** when only a subset of the data is labeled. Grounded in an information-theoretic framework, SEAL introduces two key components—Cross-Granularity Consistency and Semantic-Guided Hierarchical Soft Contrastive Learning—to provide **structure-aware supervision** across multiple levels of semantic granularity. These components enable the model to capture fine-grained semantic relationships and improve generalization to novel categories.
>
> While SEAL and DebGCD achieve comparable performance under DINO (**64.5** vs. 64.4 avg.), SEAL consistently outperforms DebGCD, and the advantage becomes more pronounced under the stronger DINOv2 backbone (**76.3** vs. 74.9). This suggests that while DebGCD is highly effective at correcting label bias under *weak initialization*, SEAL excels in utilizing **hierarchical structure**, especially when the feature space is less noisy.
>
> Furthermore, we perform a direct comparison of model complexity in terms of model parameters and FLOPs, and find that SEAL introduces significantly fewer parameters and lower computational overhead, making it a more **lightweight** solution.
>
> |Method|#Params (MB) ↓ |||GFLOPs ↓ || |
> |-|-|-|-|-|-|-|
> ||SCars|CUB|Aircraft| SCars|CUB|Aircraft   |
> |DebGCD|705.1|704.4|705.1| 17.6| 17.6|17.6|
> |Ours|660.5| 688.0 | 688.1| 17.6|17.6|17.6|
>
>
> **Q2&W2 Performance on CUB**
>
> Thanks for this insightful question. We attribute the relatively lower performance gain on CUB not to a limitation of our method, but rather to the **unique data distribution** of the CUB dataset, which presents specific challenges that affect certain types of label assignment strategies.
>
> Specifically, we observe that CUB exhibits a higher degree of **intra-class multi-modality**—that is, individual classes tend to contain **multiple distinct sub-clusters**—compared to other datasets such as Scars and Aircraft. To empirically validate this, we conduct an analysis using Gaussian Mixture Models (GMMs) on PCA-reduced, L2-normalized features, selecting the optimal number of components M within each class via BIC (Such operation is standard for mixture selection [a], used in [b] to estimate the multimodality of a given distribution and used in class‑number estimation [c]).
> Our results show that **145 out of 200** classes in CUB require $M\geq2$, indicating multi-modal structure, whereas only 7 out of 196 in SCARS and 20 out of 100 in Aircraft show such behavior. This confirms that CUB classes are often fragmented into multiple sub-modes. We report the average number of components per class and the percentage of classes with $M\geq2$.
>
> |CUB||Aircrafts||Scars||
> |-|-|-|-|-|-|
> |Avg.M|% ($M\geq2$)|Avg.M|% ($M\geq2$)|Avg.M|% ($M\geq2$)|
> |**2.07**|**72.5**|1.2|20|1.04|3.57|
>
> Under such conditions, non-parametric clustering method like k-means—as used in Infosieve [41]—are better suited to capture these sub-clusters. In contrast, our method uses a parametric classifier trained with cross-entropy, following the widely adopted SimGCD [57] framework. While this design is effective in many settings, it may be structurally under-specified for modeling multi-modal distributions, leading to suboptimal label assignment on CUB.
>
> To further examine this, we re-evaluate our model using the semi-supervised k-means in [41] clustering (averaged over 3 runs) on extracted features. Under this semi-supervised k-means setting, our method outperforms Infosieve [41] on CUB, especially when using the strong DINOv2 backbone, which reduces initialization noise and yields more structured features. Additionally, applying this to the SimGCD baseline yields a similar performance boost, further confirming that the lower performance is not due to the learned representations, but rather due to a mismatch between clustering method and the underlying data structure.
>
> We will clarify this in the final version and include the k-means evaluation results to present a more complete picture of our method’s effectiveness.
>
> [a] Bishop, Pattern Recognition and Machine Learning, Sec. 9 (2006).
>
> [b] Wasserstein Distances, Neuronal Entanglement, and Sparsity, ICLR(2025)
>
> [c] X‑means: Extending k‑means with efficient estimation of the number of clusters, ICML(2000).
>
> | Method||CUB||
> |-|-|-|-|
> ||**All**|**Old**| **New** |
> |CiPR| 78.3 | 73.4  | 80.8 |
> |SimGCD (Reproduce)|72.3| 74.1| 71.4    |
> |SimGCD + KMeans (3runs)| 76.8$\pm$0.4  | 76.4$\pm$1.0     | 77.1$\pm$0.3    |
> |InfoSieve(Reproduce) |80.7| 84.5    | 78.8 |
> |DebGCD  |77.5| 80.8| 75.8 |
> |SEAL| 76.7| 78.3 | 75.0|
> |SEAL + KMeans (3runs)| **81.2$\pm$0.1**| **79.8$\pm$0.4** | **81.9$\pm$0.3**|
>
> **Q3 Difference with related work**
>
> Thanks for the helpful suggestion. We will revise the presentation accordingly with a more detailed discussion.
>
> **Key Advantages of Our Hierarchical Semantic-Guided Approach** :
>
> 1. **Semantic-Guided Hierarchical Supervision**
>    Unlike Infosieve [41], which encodes category structure using binary paths in an implicit decision tree (thus assuming a fixed latent tree structure), and TIDA [55], which relies on handcrafted artificial hierarchies, which may introduce erroneous supervision as evidenced by the performance gap between TIDA and SEAL (e.g., **54.7 vs. 66.2** on CUB, **54.6 vs. 62.0** on Aircraft). SEAL is the first to incorporate **semantically guided** hierarchical labels into the learning process. These labels are readily available in many real-world scenarios and provide meaningful structural cues.
> 2. **Semantic-Aware Soft Contrastive Learning**
>    Prior contrastive learning methods (e.g., GCD [48], SimGCD [57], DebGCD [33]) treat all negative samples equally. In contrast, SEAL introduces **semantic-aware soft contrastive learning** by leveraging the class hierarchy to impose graded penalties.
>    For instance, samples from semantically similar categories—such as **Siamese** and **Egyptian Mau** (both domestic cats)—should be penalized less than distant ones like **Beagle** (a dog breed).
>    This structure-aware contrastive learning improves representation quality and allows more nuanced inter-class separation.
>
> Methods like DTC [23], RS [21, 22], UNO [18], NCL [65], DualRS [63], and JOINT [27] operate under the *Novel Category Discovery* setting, assuming all unlabelled samples are from unseen classes.  These methods leverage clustering, rank statistics, and label balancing, but are not directly applicable to GCD.
>
> GCD [48] generalizes NCD by assuming that unlabelled data contains both seen and novel classes. Under this setting, ORCA [5] introduces an uncertainty-adaptive margin to dynamically balance the learning speed between seen and novel categories. CiPR [24] constructs pseudo-labels from confident unlabelled samples through clustering and enforces consistency via a contrastive loss. DCCL [39] further improves pseudo-label quality by leveraging local structure mining in a teacher-student contrastive learning framework. SimGCD [57] establishes a strong end-to-end baseline by jointly training a feature extractor and a parametric classifier using self-distillation and entropy regularization, while SPTNet [52] enhances SimGCD with spatial prompt tuning to emphasize discriminative object regions. Infosieve [41] encodes each sample using binary paths in an implicit decision tree, which assumes a fixed latent hierarchical structure. TIDA [55], on the other hand, introduces handcrafted abstract hierarchies by constructing prototypes at manually defined levels.
>
> **W3 Backbone choose for Tab.3**
>
> Thanks for pointing this out, and we will fix the order typo of Tab.2 and Tab.3 in the final version. For Tab.3 (Pets and Herbarium), we report results using DINO, which is a common practice in the GCD community for these datasets in prior works [41,48,52,57]. To further address your concern, we have additionally included the results of our method using DINOv2 for reference, where SEAL achieves consistent improvements when switching from DINO to DINOv2, especially on the more challenging Herb19 dataset (**+12.1** for All), highlighting the scalability of our method under stronger visual representations.
>
> | Method| Oxford-Pet ||| Herb | ||
> |-|-|--|-|--|--|-|
> | **SEAL**|All| Old| New| All| Old| New|
> |DINO|92.9| 88.9|95.0|46.3|45.8 | 48.2 |
> |DINOv2|95.7| 96.3|95.4 | 58.4 | 57.1|59.91|
>
> **Limitations Section**:
>
> Thanks for pointing this out. We will revise the paper to move the limitations from Appendix to the main paper to include a concise summary of the most relevant limitations.

---

> > ### Comment · Reviewer_mwGr · 2025-08-02
> >
> > Thank you to the authors for the thorough rebuttal. The additional experiments and clarifications have successfully addressed all of my major concerns.
> >
> > Specifically, the new analysis on the CUB dataset, which highlights its high intra-class multi-modality, provides a convincing explanation for the initial performance gap. The follow-up experiment with k-means clustering was a particularly insightful addition that strengthens the motivation for the proposed method. I strongly encourage the authors to integrate these findings into the final manuscript, as they add significant value and context.
> >
> > Furthermore, the detailed comparison against DebGCD across different backbones, along with the computational complexity analysis, effectively clarified the novelty and specific advantages of your approach. I also appreciate the commitment to address my other minor suggestions.
> >
> > Given that my concerns have been fully resolved, and after considering the rebuttal and the other reviews, I will be raising my score.

---

> > > ### Author Response · Authors · 2025-08-02
> > >
> > > Thank you for the thoughtful follow-up. We sincerely appreciate your encouraging feedback and are glad that our clarifications and new analyses have addressed your concerns. We will integrate the suggested findings and comparisons into the final version to strengthen the manuscript. Thank you again for your constructive comments.

---

### Official Review · Reviewer_8DjF · 2025-07-03

**Clarity:** 3
**Significance:** 2
**Originality:** 2
**Rating:** 3
**Confidence:** 3

**Summary:**

This paper proposes the Semantic-aware Hierarchical Learning framework (SEAL), marking the first utilization of naturally biological classification system to guide Generalized Category Discovery (GCD) tasks. It addresses the limitations of existing approaches that rely on manually designed abstract hierarchies through three components: Semantic-aware Hierarchical Learning, Cross-Granularity Consistency Self Distillation, and Hierarchical Semantic-guided Soft Contrastive Learning.

**Questions:**

See Major Weaknesses

**Ethical Concerns:**

["NO or VERY MINOR ethics concerns only"]

**Limitations:**

yes

**Paper Formatting Concerns:**

There are no formatting issues in the paper.

**Quality:**

2

**Strengths And Weaknesses:**

**Paper Strengths：**

1.	The first framework to introduce biological classification systems into GCD, replacing error-prone manually designed abstract hierarchies.
2.	Moreover, compared with the similar work DebGCD, the effect is significantly better.
3.	This work will have a good effect in data with a clear natural hierarchy
4.	The ablation experiment is sufficient.

**Major Weaknesses:**

1.	Static Hierarchy Dependency:
The reliance of static biological classification on predefined semantics limits its adaptability to the open world

2.	Label Sensitivity:
I notice that: 'When the real semantic structure conflicts with the preset hierarchy, performance drops sharply (e.g., non-standard biological classification)'. This also constitutes a significant limitation.

*Minor Weaknesses:*

I would recommend beautifying the images in the article.

---

> ### Author Rebuttal · Authors · 2025-07-26
>
> We sincerely thank the reviewer for the thoughtful and positive feedback. We greatly appreciate the reviewer's recognition of the value  of  **first introducing biological classification systems into GCD**,  **clear advantage over similar work DebGCD**, and **sufficient ablation studies**. Our detailed responses to the listed concerns are provided below. We will also further strengthen the final manuscript based on the reviewers' concluding feedback. A well-documented code together with all trained models will be made public.
>
> **W1 Static Hierarchy Dependency**
>
> We would like to clarify that the semantic hierarchies leveraged in our framework are **adaptable** and **well-aligned with real-world structures**, rather than rigid or artificially imposed.
> In biological sciences, hierarchical classification arises naturally and serves as a foundational principle in taxonomy. For example, in botanical research, taxonomists commonly use labelled specimens of known species to classify newly collected, unlabelled samples into existing taxonomic hierarchies or to identify novel species [a,b].
>
> Similarly, in our work, we assume that hierarchical labels are available **ONLY** for the seen categories, which is a widely adopted and realistic assumption. Such labels are commonly used and readily available in many prior studies[9,15,40]. In practice, they can be obtained directly from curated resources (e.g., the GBIF botanical database [Supp.9]) or automatically generated using large language models (LLMs), as shown in our supplementary material B.1. This design not only aligns with the natural structure of biological classification systems but also enhances the interpretability of open-world learning.
>
> To further demonstrate the practicality of our setting, we test the use of LLM-generated hierarchical labels. Specifically, we apply ChatGPT to generate taxonomic structures, and observe that the outputs align with the ground-truth hierarchies. This shows that modern LLMs are capable of producing reliable semantic hierarchies with minimal effort, and our method still achieves strong performance under this setup, validating the effectiveness and accessibility of such automatically constructed hierarchies in real-world discovery scenarios.
>
> | Method       | CUB         |             |             | SCars |             |             | Aircraft |             |             |
> |--------------|-------------|-------------|-------------|-------------|-------------|-------------|-------------|-------------|-------------|
> |              | All         | Old         | New         | All         | Old         | New         | All         | Old         | New         |
> | SEAL(LLM-generated hierarchies)     | 66.2        | 72.1       | 63.2        | 65.3          | 79.3       | 58.5        | 62.0           | 65.3        | 60.4        |
>
>
> [a] *The role of taxonomy in species conservation* - 2004
>
> [b] *Species delimitation 4.0: integrative taxonomy meets artificial intelligence* - 2024
>
> **W2 Label Sensitivity**
>
> Thanks for pointing out label sensitivity. While we would like to clarify that we do *not* state that performance drops sharply when the real semantic structure conflicts with a preset hierarchy. Rather, our concern is that incorrect or arbitrarily defined hierarchies such as the abstract superclasses manually designed in prior work [55] may introduce *misleading supervisory signals* due to their lack of semantic grounding.
>
> To further support this point, we provide an additional experiment using randomly generated hierarchical labels, which indeed resulted in performance degradation. This highlights the importance of **semantically meaningful hierarchies** rather than abstract self-defined hierarchies.
>
> | Method       | CUB         |             |             | SCars |             |             | Aircraft |             |             |
> |--------------|-------------|-------------|-------------|---------------|-------------|-------------|----------------|-------------|-------------|
> |              | All         | Old         | New         |       All           | Old         | New         | All            | Old         | New         |
> | 100%  Random   | 30.3        | 31.6      | 29.7        | 29.6          | 31.4       | 28.7        | 33.2           | 31.3        | 34.2       |
> | 50%  Random     | 51.2        | 50.3       | 51.7        | 48.5         | 50.1       | 47.7        | 40.7           | 39.6        | 41.3        |
> | SEAL     | **66.2**        | **72.1**       | **63.2**        | **65.3**          | **79.3**       | **58.5**        | **62.0**           | **65.3**        | **60.4**        |
>
> **W3 Recommend beautifying the images in the article**
>
> Thanks for the suggestion. We will try to further beautify the images. If more detailed suggestions could be provided, we would be happy to adopt.

---

### Author Response · Authors · 2025-08-04
**General Summary**

We sincerely thank all reviewers for their careful readings, insightful comments, and positive feedback. Reviewers agreed that our paper is **well-written and easy to follow** (Reviewer mwGr) and **well-organized and well-presented** (Reviewer PVY1). Additionally, reviewers commented that our motivation is **clear and compelling, grounded in information-theoretic principles** (Reviewer mwGr), that we are the **first framework to introduce biological classification systems into GCD** (Reviewer 8DjF), and that **incorporating hierarchical information into GCD** is an **interesting direction** (Reviewer 9SBL). Furthermore, reviewers noted **better performance** (Reviewers 8DjF, PVY1, 9SBL), **sufficient ablation studies** (Reviewers 8DjF, PVY1), and a **valuable analysis of computational costs** (Reviewer mwGr).

We have carefully addressed all concerns raised by the reviewers with additional experiments and explanations. If there are any remaining concerns or additional suggestions, we would be grateful to discuss and refine further. Thanks to all reviewers for their constructive reviews. We are looking forward to the reviewers' response and further discussion.

---

### Note · Authors · 2025-08-13

We sincerely appreciate all ACs and reviewers for the time and care devoted to our submission.

We appreciate reviewers for their recognition across motivation&novelty, clarity, and empirical validation. On motivation and novelty, reviewers highlighted that ours is the **first framework** to introduce biological classification systems into GCD (Reviewer 8DjF), that the motivation is **clear and compelling** and grounded in **information-theoretic** principles (Reviewer mwGr), and that incorporating hierarchical information into GCD is an **interesting direction** aligned with **real-world semantics** (Reviewer 9SBL). On writing clarity, the paper was described as **well-written and easy to follow** (Reviewer mwGr) and **well-organized** and **well-presented** (Reviewer PVY1). Regarding experiments, reviewers pointed to **better performance** (Reviewers 8DjF, PVY1, 9SBL), **sufficient ablation studies** (Reviewers 8DjF, PVY1), and a **valuable analysis of computational costs** (Reviewer mwGr).

In our rebuttal, we addressed concerns with targeted analyses and new experiments. For differences from prior methods (Reviewers mwGr, PVY1), we clarified that SEAL is the **first** to use **semantic hierarchies** for GCD and showed, through additional results, that it is not only more effective but also more **efficient** . For the CUB performance concern (Reviewers mwGr, PVY1), we provided diagnostic experiments demonstrating that our learned representations are strong and that the observed gap stems from a mismatch between the label assignment protocol and the **dataset’s multi-modal structure**, a point that Reviewer mwGr explicitly **acknowledged**.For hierarchy label availability (Reviewers 8DjF, 9SBL, PVY1), we evaluated **LLM-generated hierarchies** along **alternative semantic dimensions** and found SEAL remains **robust**, supporting our view that semantic hierarchies are more **reliable** than abstract, hand-crafted ones used in some prior work. Finally, we reiterate that our hierarchies are derived **only** from **labeled data**, an assumption that is **common in related areas** (e.g., image classification[9,15],open-set recognition[31,58]) and mirrors real-world practice in taxonomy, where labeled specimens guide the placement of new, unlabeled samples into existing hierarchies.

Thank you again for your time, thoughtful consideration, and the reviewers’ constructive input—we sincerely appreciate it. We will reflect all the points in our revision.

---

### Decision · Program_Chairs · 2025-09-17

**Decision:**

Accept (poster)

**Comment:**

The paper proposes SEAL, a semantic-aware hierarchical learning framework for Generalized Category Discovery (GCD). SEAL is motivated by the observation that prior work often relies on single-level semantics or manually defined abstract hierarchies, which may be error-prone. Instead, SEAL leverages naturally occurring semantic hierarchies (e.g., biological taxonomies) to guide learning. The framework combines Hierarchical Semantic-Guided Soft Contrastive Learning, which improves negative sampling, with Cross-Granularity Consistency, which aligns predictions across coarse and fine levels. Extensive experiments on multiple benchmarks show strong results, with ablations and visualizations highlighting the contributions of each component.

Reviewers acknowledged several contributions: SEAL is the first framework to explicitly introduce biological classification systems into GCD (R1), its motivation is “clear and compelling” and theoretically grounded (R2), and incorporating hierarchical information aligns well with real-world semantics (R3). The paper was described as well-written and well-organized (R2, R4), and the experimental evaluation was supported by sufficient ablation studies and computational analysis (R1, R2, R4). SEAL shows competitive or state-of-the-art results on fine-grained datasets, outperforming methods such as DebGCD and TIDA in several cases (R1, R4).

Concerns were raised about clarity (R3), the assumption of fixed hierarchies and class counts (R1, R3, R4), and novelty relative to prior works such as DebGCD and LegoGCD (R2, R4). Performance inconsistencies (e.g., lower accuracy on CUB) were also noted (R2, R4). However, the authors clarified that hierarchies are derived only from labeled data, a practice consistent with real-world taxonomy and with common assumptions in related work (e.g., SimGCD, InfoSieve, DebGCD).  They also provided strong new experiments showing SEAL’s robustness under LLM-generated hierarchies, random hierarchies, and scenarios without known class counts. These additional results convincingly addressed the realism of assumptions and demonstrated that SEAL is not tied to a single predefined hierarchy, and that semantic-guided hierarchies -- rather than abstract, hand-crafted ones -- are key for GCD, whether sourced from curated taxonomies, generated by LLMs, or defined along different semantic dimensions.  Given these clarifications, the main reservations about applicability and realism are resolved.

The paper received final ratings of 2x borderline rejects (R1, R3) and 2x borderline accepts (R2, R4).  While initial consensus leaned negative, the authors’ thorough rebuttal -- including results on random/LLM-generated hierarchies and clarification of practical assumptions -- demonstrated that SEAL is both realistic and impactful.  Since natural hierarchies are abundant in real-world applications, SEAL makes a timely and meaningful contribution to GCD research.